# Novel and optimized mouse behavior enabled by fully autonomous HABITS: Home-cage assisted behavioral innovation and testing system

**Bowen Yu**[1,2,3], **Penghai Li**[1,2,3], **Haoze Xu**[1,2,3], **Yueming Wang**[2,4], **Kedi Xu**[2,3], **Yaoyao Hao**[1,2,4]*

[1]The State Key Lab of Brain-Machine Intelligence, Zhejiang University, Hangzhou, China; [2]Nanhu Brain-computer Interface Institute, Hangzhou, China; [3]Department of Biomedical Engineering, Zhejiang University, Hangzhou, China; [4]College of Computer Science and Technology, Zhejiang University, Hangzhou, China

## eLife Assessment

This manuscript describes a novel approach for assessing cognitive function in freely moving mice in their home-cage, without human involvement. The authors provide **convincing** evidence in support of the tasks they developed to capture a variety of complex behaviors and demonstrate the utility of a machine learning approach to expedite the acquisition of task demands. This work is **important** given its potential utility for other investigators interested in studying mouse cognition.

*For correspondence:
yaoyaoh@zju.edu.cn

**Abstract** Mice are among the most prevalent animal models used in neuroscience, benefiting from the extensive physiological, imaging, and genetic tools available to study their brain. However, the development of novel and optimized behavioral paradigms for mice has been laborious and inconsistent, impeding the investigation of complex cognitions. Here, we present a home-cage assisted mouse behavioral innovation and testing system (HABITS), enabling free-moving mice to learn challenging cognitive behaviors in their home-cage without any human involvement. Supported by the general programming framework, we have not only replicated established paradigms in current neuroscience research but also developed novel paradigms previously unexplored in mice, resulting in more than 300 mice demonstrated in various cognition functions. Most significantly, HABITS incorporates a machine-teaching algorithm, which comprehensively optimized the presentation of stimuli and modalities for trials, leading to more efficient training and higher-quality behavioral outcomes. To our knowledge, this is the first instance where mouse behavior has been systematically optimized by an algorithmic approach. Altogether, our results open a new avenue for mouse behavioral innovation and optimization, which directly facilitates investigation of neural circuits for novel cognitions with mice.

## Introduction

Complex goal-directed behaviors are the macroscopic manifestation of high-dimensional neural activity, making animal training in well-defined tasks a cornerstone of neuroscience (*Carandini, 2012*; *Krakauer et al., 2017*; *Niv, 2021*). Over the past decades, researchers have developed a diverse array of cognitive tasks and achieved significant insights into the underlying cognitive computations (*Nau et al., 2024*). Mouse has been increasingly utilized as model animal for investigating neural

**eLife digest** Mice are widely used in neuroscience research due to the many tools available to study their brain function and behavior. However, training mice for complex tasks requires extensive human involvement, which can stress the animals and introduce inconsistencies in methods and results.

Automated systems can reduce any potential bias, but most focus on single tasks only and lack optimization. To address these issues, Yu et al. developed the Home-cage Assisted Behavioral Innovation and Testing System (HABITS) – a fully autonomous platform where mice learn tasks in their cages without human intervention.

Using HABITS, mice successfully acquired a wide range of cognitive skills – including decision-making, working memory, and attention – entirely without handling. The system uses machine learning to adjust training sequences, improving learning speed and minimizing bias. In tests with over 300 mice across more than 20 paradigms, including some never attempted in mice, HABITS also improves the overall health of mice compared to the conventionally used water-restriction training. The system's AI-driven adjustments help mice learn challenging tasks more efficiently and with fewer errors.

Its low cost and automation make it an efficient and reliable tool to study behavior, making it suitable for large-scale studies. Future developments may incorporate wireless neural recordings to directly link behavior with brain activity, providing deeper insight into learning and decision-making mechanisms.

mechanisms of decision making, due to the abundant tools available in monitoring and manipulating individual neurons in intact brain (*Vázquez-Guardado et al., 2020*). One notable example is the field of motor planning *Tanji and Evarts, 1976*, which, after introduced into mouse models, was significantly advanced by obtaining causal results from whole brain circuits to genetics (*Svoboda and Li, 2018*).

Traditionally, training mice in cognitive tasks was inseparable from human involvement in frequent handling and modifying shaping strategies according to the performance, thus labor-intensive and inconsistent as well as introducing unnecessary noise and stress (*Balcombe et al., 2004*). Recently, many works dedicated to design standard training setups and workflows, aiming for more stable and reproducible outcomes (*Findling et al., 2023*; *Han et al., 2018*; *Benson et al., 2023*; *Mah et al., 2023*; *Mohammadi et al., 2024*; *Pan-Vazquez et al., 2024*; *Rich et al., 2024*; *Scott et al., 2013*; *Aguillon-Rodriguez et al., 2021*; *Zhou et al., 2024*). For example, the International Brain Laboratory has recently shown mice can perform the task comparably across labs after a standard training within identical experimental setup (*Aguillon-Rodriguez et al., 2021*). However, human intervention still has been a significant factor, which inevitably introduced the variability. Furthermore, training efficiency was still restricted by the limited experimental time as before, which necessitates motivational techniques like water restriction. The training sessions were often restricted to a specific task and have not been extensively tested in other complex paradigms because of the requirement of prolonged training duration. These limitations highlighted the challenges to broadly and swiftly study comprehensive cognitive behaviors in mice.

A promising solution to this scenario is to implement fully autonomous training systems. In recent years, researchers have focused on combining home-cage environments with automated behavioral training methodologies, offering a viable avenue to realize autonomous training (*Aoki et al., 2017*; *Bernhard et al., 2020*; *Bollu et al., 2019*; *Caglayan et al., 2021*; *Francis et al., 2019*; *Francis and Kanold, 2017*; *Hao et al., 2021*; *Kiryk et al., 2020*; *Murphy et al., 2020*; *Murphy et al., 2016*; *Poddar et al., 2013*; *Qiao, 2019*; *Salameh et al., 2020*; *Silasi et al., 2018*). For instance, efforts have been made to incorporate voluntary head fixation within the home-cage to train mice on cognitive tasks (*Aoki et al., 2017*; *Hao et al., 2021*; *Murphy et al., 2020*; *Murphy et al., 2016*). At the cost of increased training difficulty, they successfully integrate large-scale whole-brain calcium imaging (*Aoki et al., 2017*; *Murphy et al., 2020*; *Murphy et al., 2016*) and optogenetic modulations *Hao et al., 2021* with fully automated home-cage behavior training. Moreover, other groups have conducted mouse behavior training in group-housed home-cages and utilize RFID technology to distinguish

individual mice (*Kiryk et al., 2020*; *Murphy et al., 2020*; *Qiao, 2019Qiao, 2019*; *Silasi et al., 2018*). There are also studies which directly train freely moving animals in their home-cage to reduce stress and facilitate deployment (*Balcombe et al., 2004*; *Francis et al., 2019*; *Kiryk et al., 2020*; *Poddar et al., 2013*; *Qiao, 2019Qiao, 2019*; *Silasi et al., 2018*; *Jankowski et al., 2023*; *Schaefer and Claridge-Chang, 2012*). However, many of these studies have focused on single paradigms and incorporated complex components in their systems, which hindered high-throughput deployment for high-efficiency and long-term behavioral testing and exploring.

The training protocols employed in existing studies, no matter in manual or autonomous training, were often artificially designed (*Aguillon-Rodriguez et al., 2021Aguillon-Rodriguez et al., 2021*; *Hao et al., 2021*), potentially failing to achieve optimal outcomes. For instance, a key issue in cognitive behavioral training is that the mouse is likely to develop bias, that is obtaining reward only from one side. Various 'anti-bias' techniques *Aguillon-Rodriguez et al., 2021*; *Hao et al., 2021*; *Do et al., 2023* have been implemented to counteract the bias, yet their efficacy in accelerating training or enhancing testing reliability remains unproven. From a machine learning standpoint, if we can accurately infer the animal's internal models, it is possible to select a specific sequence of stimuli that will reduce the 'teaching' dimension of the animal and thus maximize the learning rate (*Zhu et al., 2018*). Recently, an adaptive optimal training policy, known as AlignMax, was developed to generate an optimal sequence of stimuli to expedite the animal's training process in simulation experiments (*Bak et al., 2016*). While many relative works have realized the theoretical demonstration of the validity of machine teaching algorithms under specific conditions, these have been limited to teaching silicon-based learners in simulated environments (*Bak et al., 2016*; *Jha et al., 2024*; *Liu et al., 2017*). The direct application and empirical demonstration of these algorithms in real-world scenarios, particularly in teaching animals to master complex cognitive tasks, remains unexplored. There are two fundamental barriers for testing these algorithms in real animals training: firstly, traditional session-based behavioral training results in a discontinuous training process, introducing abrupt variation of learning rate and uncontrollable noise (*Aguillon-Rodriguez et al., 2021*; *Roy et al., 2021*), which could undermine the algorithm's capability; secondly, the high computation complexity of model fitting (*Bak et al., 2016*; *Jha et al., 2024*) pose challenges for deployment on microprocessors, thereby impeding extensive and high-throughput experiments. In the lack of human supervision, a fully autonomous behavioral training system necessitates an optimized training algorithm. Therefore, integrating fully automated training with machine teaching-based algorithms could yield mutually beneficial outcomes.

To address these challenges, we introduced a home-cage assisted behavioral innovation and testing system, referred to as HABITS, which is a comprehensive platform featuring adaptable hardware and a universal programming framework. This system facilitates the creation of a wide array of mouse behavioral paradigms, regardless of complexity. With HABITS, we have not only replicated established paradigms commonly used in contemporary neuroscience research but have also pioneered novel paradigms that have never been explored in mice. Crucially, we have integrated a machine teaching-based optimization algorithm into HABITS, which significantly enhances the efficiency of both training and testing. Consequently, this study provides a holistic system and workflow for a variety of complex, long-term mouse behavioral experiments, which has the potential to greatly expand the behavioral reservoir for neuroscience and pathology research.

## Results
### System design of HABITS
The entire architecture of HABITS was comprised of two parts: a custom home-cage and behavioral components embedded in the home-cage (*Figure 1A*). The home-cage was made of acrylic plates with a dimension of 20×20 × 30 cm, which is more extensive than most of the commercial counterparts for single-housed mice. A tray was located at the bottom of the cage where ad libitum food, bedding, nesting material (cotton), and enrichment (a tube) were placed (*Figure 1B*). Experimenters can change the bedding effortlessly just by exchanging the tray. The home-cage also included an elevated, arc-shaped weighting platform inside, providing a loose body constraint for the mouse during task performing (*Figure 1A*). Notably, a micro load cell was installed beneath the platform, which can read out body weight of the mouse for long-term health monitoring. The cage was compatible with the standard mouse rack and occupied as small a space as standard mouse cage (*Figure 1C*).

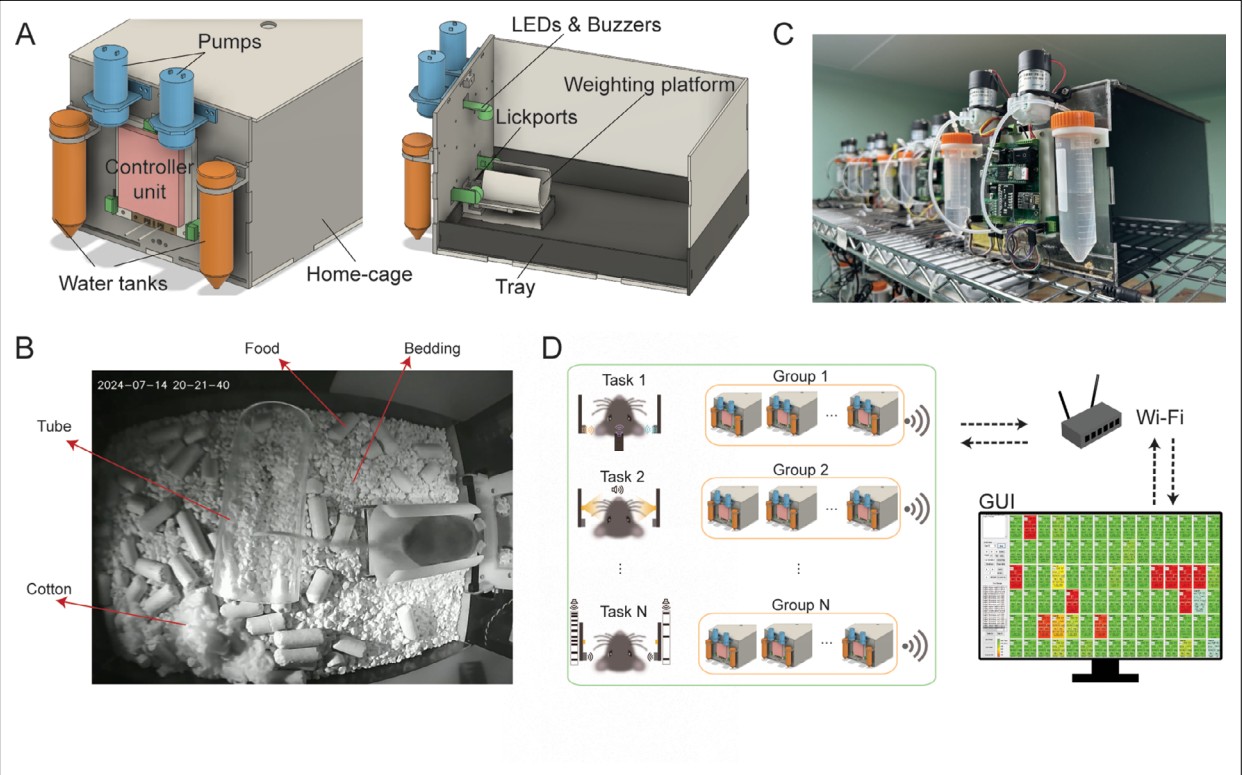

**Figure 1.** System setup for HABITS. (**A**), Front (left) and side (right) view of HABITS, showing components for stimulus presenting (LEDs & buzzers), rewarding (water tanks and pumps), behavioral reporting (lickports) and health monitoring (weight platform). These components are coordinated by the controller unit and integrated into the mouse home-cage with a tray for bedding change. (**B**), HABITS installed on standard mouse cage rack. (**C**), Mouse, living in home-cage with food, bedding, nesting material (cotton), and enrichment (tube), is performing task on the weight platform. (**D**), System architecture for high-throughput behavioral training, showing different tasks are running in parallel groups of HABITS, which further wirelessly connect to one single PC through Wi-Fi to stream real-time data to the graphic user interface (GUI).

The online version of this article includes the following figure supplement(s) for figure 1:

**Figure supplement 1.** HABITS system.

To perform behavioral training and testing in HABITS, we constructed a group of training-related components embedded in the front panel of the home-cage (*Figure 1A*). Firstly, three stimulus modules (LEDs and buzzers) for light and sound presenting were protruded from the front panel and placed in the left, right, and top positions around the weighting platform, enabling generation of visual and auditory stimulus modalities in three different spatial locations. The mouse reported the decisions about the stimuli by licking either left, right, or middle lickports installed in the front of the weighting platform. Finally, peristaltic pumps draw water from water tanks into lickports, serving as the reward for the task, which was the sole water source for the mouse throughout the period living in the home-cage. In the most common scenario, mice living in home-cage stepped on the weighting platform and triggered trials by licking on the lickports to obtain water (*Figure 1B*, *Video 1*).

To endow autonomy to HABITS, a microcontroller was used to interface with all training-related components and coordinate the training procedure (*Figure 1—figure supplement*

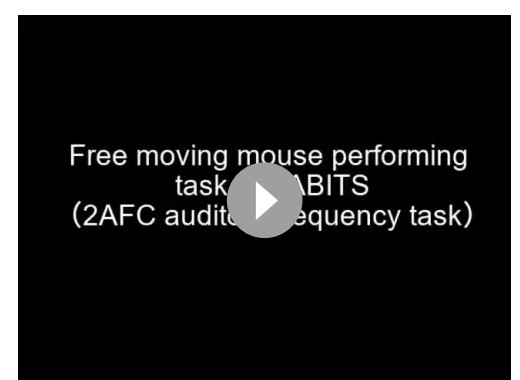

**Video 1.** Free-moving mouse performing task in HABITS.
https://elifesciences.org/articles/104833/figures#video1

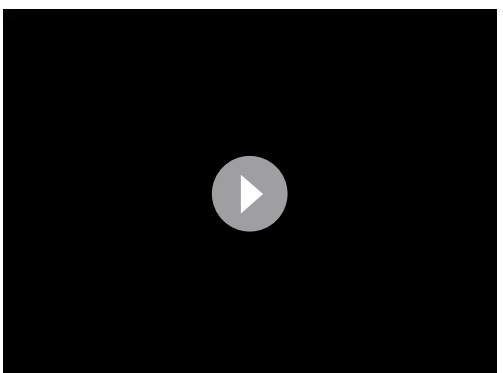

**Video 2.** The 24 hr activities of mice living in HABITS. https://elifesciences.org/articles/104833/figures#video2

1A). We implemented a microcontroller-based general programming framework to run finite state machine with millisecond-level precision (see Materials and methods). By using the APIs provided by the framework, we can easily construct arbitrarily complex behavioral paradigms and deploy them into HABITS (*Figure 1—figure supplement 1B*). Meanwhile, the paradigms were usually divided into small steps from easy to hard and advanced according to the performance of the mouse. Another important role of the microcontroller was to connect the HABITS to PC wirelessly and stream real-time behavioral data via a Wi-Fi module. A graphic user interface (GUI), designed to remotely monitor each individual mouse's performance, displayed history performance, real-time trial outcomes, body weights, etc. (*Figure 1—figure supplement 1C*). Meanwhile, the program running in the microcontroller could be updated remotely by the GUI when changing mouse and/or task. As a backup, information of training setup, task parameters, and all behavioral data were also stored in an SD card for offline analysis.

To increase the throughput of behavioral testing, we built more than a hundred independent HABITS and installed them on standard mouse racks (*Figure 1—figure supplement 1D*). The total material cost for each HABITS was less than $100 (*Supplementary file 1*). All HABITS, operating different behavioral tasks across different cohorts of mice, were organized into groups according to the tasks and wirelessly connected to a single PC (*Figure 1D*). The states of each individual HABITS can be accessed and controlled remotely by monitoring the corresponding panels in the GUI, thereby significantly improving the experimental efficiency. We developed a workflow to run behavioral training and testing experiments in HABITS (*Figure 1—figure supplement 1E*). Firstly, initiate the HABITS system by preparing the home-cage, deploying training protocols, and weighing the naive mouse that did not need to undergo any water restriction. Then, the mouse interacted with fully autonomous HABITS at their own pace without any human intervention. In this study, mice housed in HABITS went through 24/7 behavioral training and testing for up to 3 months (*Video 2*), although longer duration was supported. Finally, data were harvested from the SD card and offline analyzed.

Therefore, HABITS permitted high-throughput, parallel behavioral training and testing in a fully autonomous way, which, contrasting with manual training, allows for possible behavioral innovation and underlining neural mechanism investigation.

## HABITS performance probed by multimodal d2AFC task

We next deployed a well-established mouse behavioral paradigm, delayed two-alternative forced choice (d2AFC), which was used to study motor planning in mouse *Guo et al., 2014b*; *Inagaki et al., 2018*, in HABITS to demonstrate the performance of our system (*Figure 2*).

Mice, living in HABITS all the time, licked any of the lickports to trigger a trial block at any time. In the d2AFC task with sound frequency modality (*Figure 2A*), mice needed to discriminate two tone frequencies presented at the beginning of the trial for a fixed duration (sample epoch, 1.2 s) and responded to the choice by licking left (low tone) or right (high tone) lickports following a brief 'go' cue after a fixed delay duration (1.2 s). Licks during the delay epoch were prohibited, and unwished licks (early licks) will pause the trial for a while. Correct choices were rewarded, while incorrect choices resulted in noise and timeout. If mice did not lick any of the lickports after the 'go' cue for a fixed period (i.e. no-response trial), the trial block was terminated. The next trial block was immediately entering the state of to be triggered. Mice can only learn the stimulus-action contingency by trial-and-error. To promote learning, we designed an algorithm comprised of many subprotocols to shape the behavior step by step (*Figure 2—figure supplement 1A*). *Figure 2B* illustrated example licks during correct, error, and early lick trials for the task. As training progressed, the correct rate increased and early lick rate declined gradually for the example mouse within the first 2 week (*Figure 2C*). All the

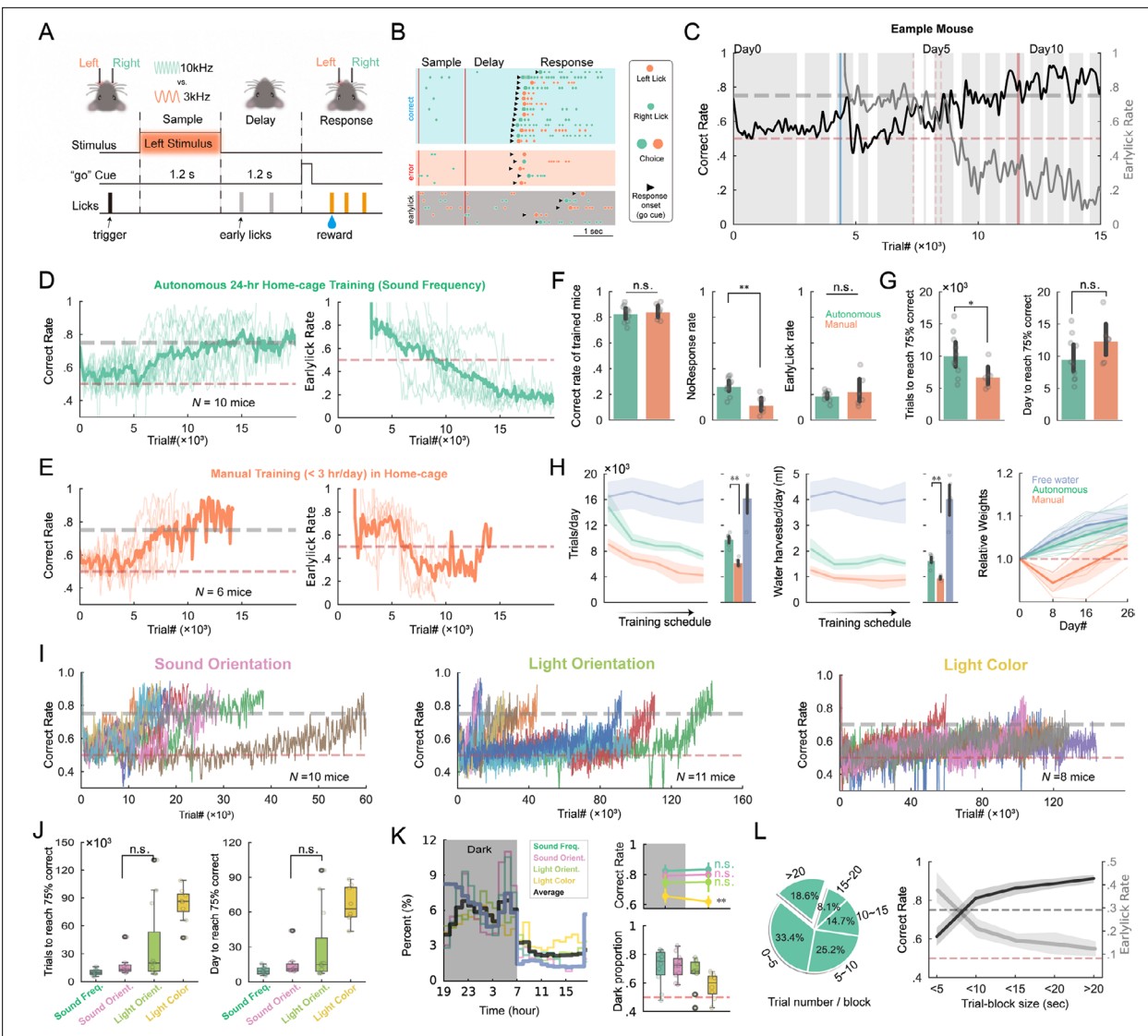

**Figure 2.** HABITS performance in d2AFC task. (**A**) Task structure for d2AFC based on sound frequency. (**B**) Example licks for correct (blue), error (red), and early lick (gray) trials. Choice is the first lick after response onset. (**C**) Correct rate (black line) and early lick rate (gray line) of an example mouse during training in HABITS for the first 13 days. Shaded blocks indicate trials occurred in the dark cycle. Trials with early lick inhabitation only occur after the blue vertical line. Red vertical dash lines represent delay duration advancement from 0.2 s to 1.2 s. (**D**) Averaged correct rate (left) and early lick rate (right) for all mice trained in d2AFC. Criterion level (75%) and chance level (50%) are labeled as gray and red dash lines, respectively. (**E**) Same as (**D**) but for manual training (1~3 hr/day in home-cage). (**F**) Averaged correct rate (left), early lick rate (middle), and no response rate (right) of expert mice trained with the two protocols. (**G**) Averaged number of trials (left) and days (right) to reach the criterion performance for the two training protocols. Circles, individual mice. Error bar, mean, and 95% CI across mice. (**H**) Left, number of trials performed per day throughout the training schedule for three different protocols. Error bar indicates the mean and 95% confidence interval (CI) across mice. Middle, volume of water harvested per day. Right, Relative body weights of mice in days 0, 8, 16, 26. Bold line and shades indicate mean and 95% CI across mice. (**I**) Behavioral performance of all mice training in d2AFC task based on sound orientation (left), light orientation (middle), and light color (right). (**J**) Box plot of average number of trials (left) and days (right) to reach the criterion performance for d2AFC tasks with different sensory modalities. (**K**), Left, percentage of trials performed as a function of time in a day for the four modalities trained autonomously (thick black shows the average). Shaded area indicates the dark cycle. Top right, averaged correct rate of grouped mice in dark cycle versus light cycle. Error bars show 95% CI across mice. Bottom right, box plot of the averaged proportion of trials performed in dark cycle for the four modalities. Data collected from expert mice. (**L**) Left, percentage of trials in blocks with varying number of consecutive trials for automated training in home-cage. Right, correct rate and early lick rate as functions of trial block size. Gray dash line, the criterion performance; Red dash line, chance performance level. Data collected from trials of expert mice. For significance levels not mentioned in all figures, n.s., not significant, p>0.05; *, p<0.05; **, p<0.01 (two-sided Wilcoxon rank-sum tests).

The online version of this article includes the following figure supplement(s) for figure 2:

**Figure supplement 1.** Autonomous versus manual training in home-cage.

*Figure 2 continued on next page*

*Figure 2 continued*

**Figure supplement 2.** Reaction-time-based 2AFC task training in home-cage automatically.

**Figure supplement 3.** Value-based dynamic foraging task.

10 mice enrolled in this task effectively learned the task and suppressed licking during the delay period within 15 days (*Figure 2D*), achieving an average of 980±25 (mean ± SEM) trials per day.

We also tested another training scheme that is limited duration per day in HABITS to simulate the situation of manual training with human supervision. These mice were water-restricted and manually transferred from traditional home-cage to HABITS daily for 1–3 hr training sessions. The duration of sessions was determined by the speed of harvesting water as we controlled the daily available volumes of water to approximately 1 ml (*Guo et al., 2014a*). All six mice learned the task as the autonomous counterpart (*Figure 2*) with similar final correct and early lick rate (except that the no-response rate was significantly lower for manual training; *Figure 2F*). Logistic regression of the mice's choice revealed similar behavioral strategies were utilized throughout the training for both groups (*Figure 2—figure supplement 1B-D*). Autonomous training needed significantly more trials than manual training (10,164 ± 1062 vs. 6845 ± 782) to reach the criterion performance; however, the number of days was slightly less due to the high trial number per day (*Figure 2G*). As shown in *Figure 2H*, autonomous training yielded significantly higher number of trial/day (980 ± 25 vs. 611 ± 26, *Figure 2H* left) and more volume of water consumption/day (1.65 ± 0.06 vs. 0.97 ± 0.03 ml, *Figure 2H* middle), which resulted in monotonic increase of body weight that was even comparable to the free water group (*Figure 2H* right). In contrast, the body weight in the manual training group experienced a sharp drop at the beginning of training and was constantly lower than the autonomous group throughout the training stage (*Figure 2H* right). As the training advanced, the number of trials triggered by mice per day decreased gradually for both groups of mice, but the water consumption per day kept relatively stable. At the end of manual training, we transferred all mice to autonomous testing and found that the number of trial and consumption water per day dramatically increased to the level of the autonomous training, suggesting mice actually needed more water throughout the day (*Figure 2—figure supplement 1E*). These results indicated that autonomous training achieved similar training performance as manual training and maintained a more healthy state of mice.

Three more cohorts of mice were used in d2AFC tasks with different modalities, which included sound orientation (left vs. right sound), light orientation (left vs. right light), and light color (red vs. blue). Using the same training protocol as in the sound frequency modality, we trained 10, 11, and 8 mice on these tasks, respectively (*Figure 2I*). Mice required different amounts of trials or days to discriminate these modalities, with light color discrimination being the most challenging (an average of 82,932±6817 trials), consistent with the limited sensitivity to light wavelength of mice (*Figure 2J*). We also tried other modalities, like blue vs. green and flashed blue vs. green, but all failed (see *Table 1*). The learning rate between sound and light orientation discrimination tasks was similar (p=0.439, two-sided Wilcoxon rank-sum test), but the variation for light orientation was large, indicating possible individual differences (*Figure 2J*). All modalities maintained good health state indicated by the body weight for up to 2 months (*Figure 2—figure supplement 1F*). Another two types of 2AFC, reaction time (*Munoz and Everling, 2004*) (RT-2AFC, *Figure 2—figure supplement 2*) and random foraging (*Figure 2—figure supplement 3*) task, were also successfully tested in HABITS. Notably, the dynamic foraging task (*Hattori et al., 2023*; *Hattori et al., 2019*), which heavily relies on historical information, was first demonstrated in a fully autonomous training scheme for freely moving mice with similar block size and performance.

Two important behavioral characteristics were revealed across all the modalities in 2AFC task. Firstly, as our high-throughput behavioral training platform operated on a 12:12 light-dark cycle, the long-term circadian rhythm of mice can be evaluated based on the number of triggered trials and performance during both cycles. We found all mice exhibited clear nocturnal behavior with peaks in trial proportion at the beginning and end of the dark period (*Figure 2K*), which was consistent with previous studies (*Francis et al., 2019*; *Hao et al., 2021*). The light color modality exhibited a slightly lower percentage of trials during dark (57.89% ± 3.18% vs. above 66% for other modalities; *Figure 2K*), possibly the light stimulus during trials affected the circadian rhythms of the mice. It was also observed that all mice except the light color modality showed no significant differences in correct rate between light and dark cycle after they learned the task (*Figure 2K*). The higher performance in a

**Table 1.** All tasks training in HABITS.

| Protocol name (abbreviation) | Modality | Animals trained (trained / used) | Note |
|---|---|---|---|
| | Sound frequency (3 k vs. 10 kHz) | 11/11 | |
| | Sound orientation (Left vs. Right) | 10/11 | |
| | Light orientation (Left vs. Right) | 10/11 | |
| | Light color (Blue VS. Red) | 8/11 | *Figure 2* |
| | Light color (Green VS. Blue) | 0/10 | Mice are insensitive to light colors. |
| delayed 2-Alternative Forced Choice (d2AFC) | Light color (flashed Green VS. Blue) | 0/10 | |
| Reaction time 2AFC (RT-2AFC) | Sound frequency (3 k vs. 12 kHz) | 6/6 | *Figure 2—figure supplement 2* |
| Contingency reversal | RT-2AFC, sound frequency (3 k vs. 12 kHz) | 8/8 | *Figure 3A* |
| Continuous learning | Sound freq. (3 k vs. 12 kHz), reversal sound freq., sound orient. (Left vs. Right), reversal sound orient., light orient. (Left vs. Right) | 10/30 | *Figure 4A*; 20 mice did not learn light oriental modal within 90 days. |
| Evidence accumulation with spatial cue | Poisson distributed clicks with spatial diff. (Left vs. Right) | 8/10 | *Figure 3C* |
| | Poisson distributed clicks and flashes (4 vs 20 events/s) | 13/15 | *Figure 3D* |
| Multimodal Integration | Poisson distributed light flashes (4 vs 20 events/s) | 3/15 | 12/15 mice failed to discriminate light flash |
| | Sound frequency (8 k vs. 32 kHz) | 5/6 | *Figure 3E* |
| Confidence probing task | Sound frequency (8 k vs. 32 kHz) and Poisson distributed clicks with spatial diff. (Left vs. Right) | 0/12 | Delay period up to 8 sec, failed |
| Value-based dynamic foraging task | No sensory cues (Block size from 500 to 100) | 6/6 | *Figure 2—figure supplement 3* |
| | Temporal regular clicks with 5 alternative rates (8, 16, 32, 64, 128 Hz) | 5/6 | *Figure 3B* |
| Working memory task | Temporal regular clicks with 3 alternative rates (8, 32, 128 Hz) | 8/8 | *Figure 3—figure supplement 1* C |
| | Sound frequency (3 k & 12 kHz) | 10/10 | *Figure 4B*; Sample and test period: 500ms |
| double Delayed Match Sample (dDMS) | Sound frequency (3 k & 12 kHz) | 0/10 | Random sample and test period: 100 or 1000ms |
| | Sound frequency (8 k vs. 16 k vs. 32 kHz) | 14/14 | *Figure 4C* |
| | Sound frequency reversal (8 k vs. 16 k vs. 32 kHz) | 4/4 | *Figure 3—figure supplement 1B* |
| delayed 3-Alternative Forced Choice (d3AFC) | Sound orientation (Left vs. Middle vs. Right) | 6/6 | *Figure 3—figure supplement 1A* |
| | Sound frequency (3 k vs. 12 kHz) or sound orientation (Left vs. Right); regular click rates (16 vs 64 Hz) as context cue | 6/6 | *Figure 4D* |
| | Sound orientation (Left vs. Right) or light orientation (Left vs. Right); regular click rates (16 vs 64 Hz) as context cue | 0/10 | |
| Context-dependent attention task | Sound orientation (Left vs. Right) or light orientation (Left vs. Right); Sound frequency (3 k vs. 12 kHz) as context cue | 0/20 | Mice failed in light modality |

*Table 1 continued on next page*

*Table 1 continued*

| Protocol name (abbreviation) | Modality | Animals trained (trained / used) | Note |
|---|---|---|---|
| | Working memory task, Temporal regular clicks with 5 alternative rates and full stimulus matrix (8, 16, 32, 64, 128 Hz) | 7/7 (MT) 5/8(Random) | *Figure 5B* |
| | RT-2AFC, Sound frequency (3 k vs. 12 kHz) | 10/10 (MT) 10/10 (random) 10/10 (antibias) | *Figure 5F*; Same group of mice used in continuous learning |
| Machine teaching algorithm | 2AFC with Sound frequency (3 k vs. 12 kHz), sound orientation (Left vs. Right) and sound orientation reversal, respectively | 10/10 (MT) 8/8 (random) | *Figure 6* |
| Total | N/A | 200/284 | N/A |

dark environment for light color modality implied that light stimulus presented in a dark environment was with higher contrast and thus better discernibility. Secondly, as we organized the trials into blocks, the training temporal dynamic at trial-level could be examined. We found more than two-thirds of the trials were done in >5-trial blocks (*Figure 2L* left), which resulted in more than 55% of the trials being with inter-trial intervals less than 2 s (*Figure 2—figure supplement 1H*). The averaged block duration was 27.64±1.73 s and mice triggered another block of trials within 60 s in more than 60% of cases. Meanwhile, we also found that the averaged correct rate increased and the early lick rate decreased as the length of the block increased (*Figure 2L* right), which suggested that mice were more engaged in the task during longer blocks.

These results showed that mice can learn and perform cognitive tasks in HABITS with various modalities in a fully autonomous way. During the training process, mice maintained good health conditions, although without any human intervention. Due to the high-efficiency training with less labor and time, it gave us an opportunity to explore and study more widespread cognitive behavioral tasks in mice.

## Various cognitive tasks demonstrated in HABITS

We next tested several representative tasks that were commonly used in the field of cognitive neuroscience to demonstrate the universality of HABITS. It is worth noting that many new features of the behavior could be explored due to the autonomy and advantages of HABITS, in terms of either quantity or quality (*Figure 3*).

Contingency reversal task was a cognition-demanding task and previously used to investigate cognitive flexibility (*Izquierdo et al., 2017*; *Figure 3A*). In the task, the contingency for reward switched without any cues once mice hit the criteria performance (*Figure 3A1*). Mice can dynamically reverse their stimulus-action contingency, though with different learning rates across individuals (*Figure 3A2*). Given the advantages of long-term autonomous training in HABITS, all mice underwent contingency reversal for an average of 52.25±16.39 times, and one mouse achieved up to 125 times in 113 days. Most of the mice (6/8) gradually decreased the number of trials to reach criterion across multiple contingency reversals representing an effect of learning to learn (*Hattori et al., 2023*; *Akrami et al., 2018*; *Figure 3A3*). Meanwhile, the average number of trials to reach criterion during the reversal was highly correlated with the trial number in the first reversal learning which represented the initial cognitive ability or the sensory sensitivity of individual mice (*Figure 3A4*).

Working memory was another important cognition that was vastly investigated using rodent model with auditory and somatosensory modalities (*Akrami et al., 2018*; *Fassihi et al., 2014*). Here, we utilized a novel modality, regular sound clicks, to implement a self-initiated working memory task, which required mice to compare two click rates separated by a random delay period (*Figure 3B1*). We initially validated that mice did employ a comparison strategy in a 3×3 stimulus generation matrix (SGM), instead of just taking the first cue as a context (*Figure 3—figure supplement 1A*). Subsequently, we expanded the paradigm's perception dynamics to 5×5 and reduced the relative perceptual contrast between two neighboring stimuli to one octave (*Figure 3B2*). HABITS enabled investigation of detailed behavioral parameters swiftly. We noticed that the discrimination ability for mice was significantly better in the higher frequency range (*Figure 3B3*), which may be caused by the different sensitivity of mice across the spectrum of regular click rate. During the testing stage, we

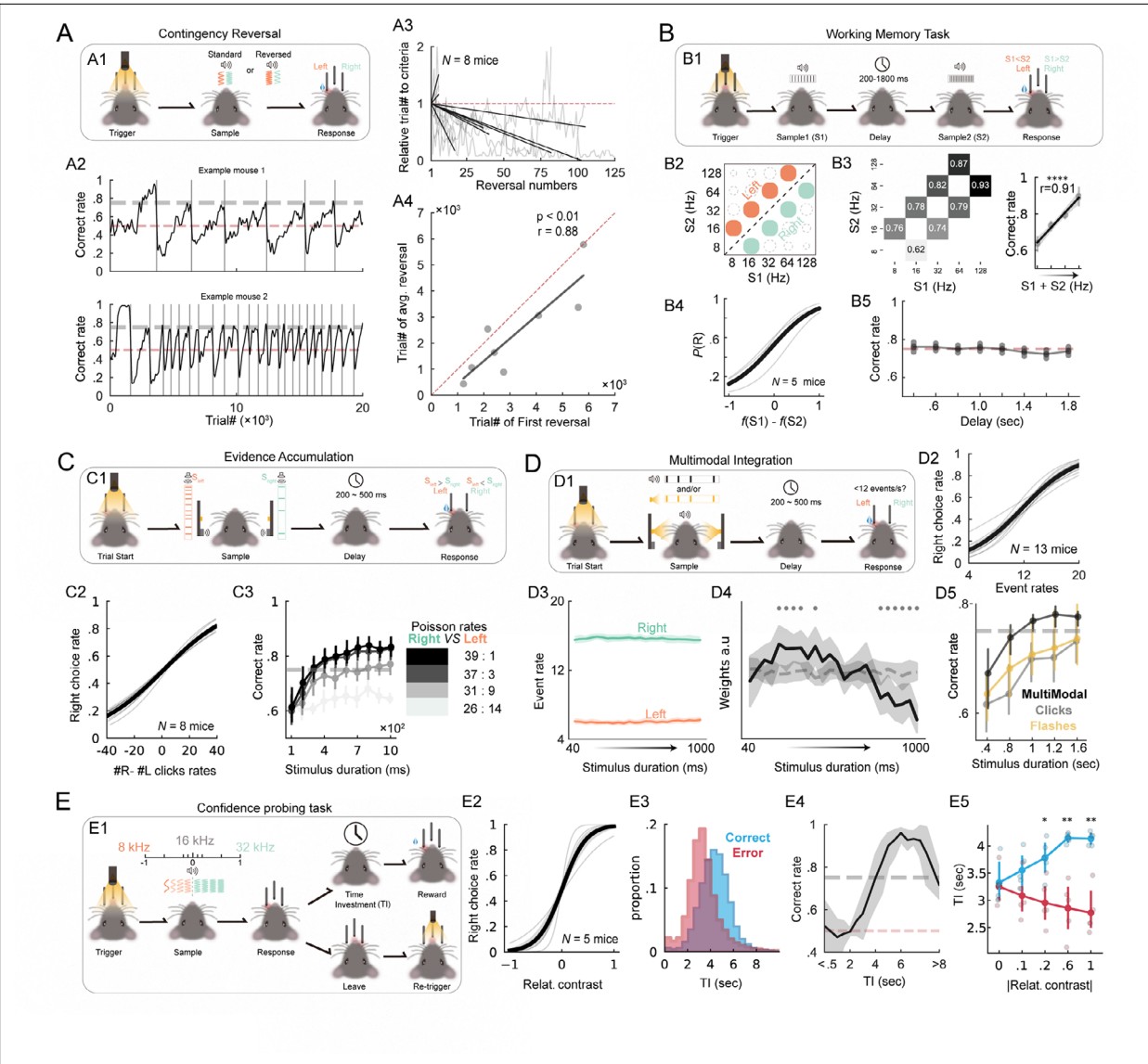

**Figure 3.** Representative cognitive task performed in HABITS. (**A**) Contingency reversal task. (**A1**) Task structure. (**A2**) Correct rate of example mice with different learning rates. Gray vertical lines indicate contingency reversal. (**A3**) Relative number of trials to reach the criterion as a function of reverse times. Gray lines, individual mice. Black lines, linear fit. (**A4**) Number of trials in the first reversal learning versus the average number of trials of the rest of contingency reversal learning for each mouse (each dot). Black line, linear regression. Red dash line, diagonal line. (**B**) Working memory task with sound frequency modality. (**B1**) Task structure. (**B2**) Stimulus generation matrix (SGM) for left (orange) and right (green) trials. (**B3**) Left, averaged correct rate for each stimulus combination tested. Right, averaged correct rate for each (S1 +S2) stimulus combination across mice. Black line and shade, linear regression and 95% CI. (**B4**) Averaged psychometric curves, that is percentage of right choice as a function of frequency difference between sample 1 and sample 2. (**B5**) Averaged correct rate as a function of delay duration. (**C**) Evidence accumulation with spatial cue task. (**C1**) Task structure. (**C2**) Averaged psychometric curves, that is performance as a function of the difference between right and left clicks rates. (**C3**) Averaged correct rate across all mice as a function of sample duration for different Poisson rates (different colors). Error bar represents 95% CI. (**D**) Multimodal integration task. (**D1**) Task structure. (**D2**) Averaged correct rate across all mice as a function of sample duration for different stimulus modalities (different colors). (**D3**) Averaged event rates during sample period for left (red) and right (blue) choice trials. (**D4**) Averaged weights (black line) of logistic regression fitting to the choice of trials across expert mice tested in >1000 trials (N=11 mice) from the first bin (40ms) to the last bin (1000ms) of the sample period. A gray dash line represents the null hypothesis. Gray dots indicate significance, p<0.05, two-sided *t*-tests. (**D5**), Psychometric curve for trials with multimodal stimulus. (**E**) Confidence probing task. (**E1**) Task structure. (**E2**) Psychometric curve, that is right choice rate as a function of relative contrast (log scaled relative frequency). (**E3**) Histogram of time invested (TI) for both correct and error trials. (**E4**) Averaged correct rate across all mice as a function of TI. (**E5**) Averaged TI as a function of absolute relative contrast for both correct and error trials. Cycles, individual mice; *, p<0.05; **, p<0.01, two-sided Wilcoxon rank-sum tests.

The online version of this article includes the following figure supplement(s) for figure 3:

**Figure supplement 1.** Other complex cognitive behavioral tasks training in home-cage automatically.

varied the contrast of the preceding stimulus while maintaining the succeeding one; the psychometric curves affirmed mice's decision-making based on perceptual comparison of two stimuli and validated their perceptual and memory capacities (*Figure 3B4*). Meanwhile, mice could maintain stable working memory during up to 1.8 s delay, demonstrating mice can perform this task robustly (*Figure 3B5*).

Evidence accumulation introduced more dynamics to the decision-making within individual trials and was widely utilized (*Brunton et al., 2013*; *Erlich et al., 2015*; *Hanks et al., 2015*). In the task implemented in HABITS, mice needed to compare the rate of sound clicks that randomly appeared over two sides during the sample epoch and made decisions following a 'go' cue after a brief delay (*Figure 3C1*). We successfully trained 8/10 mice to complete this task as revealed by the psychometric curves (*Figure 3C2*). After mice learned the task, we systematically tested the effect of evidence accumulation versus the task performance. All mice exhibited consistent behavioral patterns, which were correlated with the evidence of stimulus, that is longer sample period and/or higher stimulus contrast led to higher performance (*Figure 3C3*). It showed an evident positive correlation between evidence accumulation and task performance.

Multimodal integration *Meijer et al., 2018*; *Odoemene et al., 2018*; *Raposo et al., 2012* was also tested in HABITS with sound clicks and light flashes as dual-modality events in the evidence accumulation framework. Mice were required to differentiate whether event rates were larger or smaller than 12 event/s (*Figure 3D1*). We successfully trained 13/15 mice in this paradigm with multimodal or sound clicks stimulus (*Figure 3D2*), and only three mice achieved performance criteria in trials with light flashes stimulus. Since the events were presented non-uniformly within each trial, we wondered about the dynamics of the decision-making process along the trial. Firstly, we divided all trials with multimodal stimulus into two groups according to the choice of expert mice. The uniform distribution of events within a trial indicated that mice considered the whole sample period to make a decision (*Figure 3D3*). Secondly, we used a logistic regression model to illustrate the mice's dependency on perceptual decisions throughout the entire sample period. We found that mice indeed tended to favor earlier stimuli (i.e. higher weights) in making their final choices (*Figure 3D4*), consistent with previous research findings (*Odoemene et al., 2018*). Lastly, we further tested modulated unimodal stimuli with different sample periods in the testing stage, in which accuracy correlated positively with sample period and conditioned test for different combinations of modalities demonstrated evidence accumulation and multimodal integration effect, respectively. (*Figure 3D5*).

Confidence was another important cognition along with the process of decision-making, which was investigated in rat (*Kepecs et al., 2008*; *Masset et al., 2020*) and more recently in mice (*Schmack et al., 2021*) model previously. We introduced a confidence-probing task in HABITS (*Figure 3E*), in which mice needed to lick twice the correct side for acquiring a reward; the two licks were separated by a random delay during which licking at other lickports prematurely ended the trial (*Figure 3E1*). This unique design connects mice's confidence about the choice, which was hidden, with the time investment (TI) of mice between two licks, which was an explicit and quantitative metric. We successfully trained 5/6 mice (*Figure 3E2*), and most importantly, there was a noticeable difference of TI in correct versus incorrect trials (*Figure 3E3*). In detail, trials with longer TIs tended to have higher accuracies (*Figure 3E4*). Furthermore, the TI was also modulated by the contrasts of stimuli; as contrast decreased, mice exhibited reduced confidence about their choices, manifesting as decreased willingness to wait in correct trials, and conversely in error trials (*Figure 3E5*).

In summary, mice could undergo stable and effective long-term training in HABITS with various cognitive tasks commonly used in state-of-the-art neuroscience. These tasks running in HABITS were demonstrated to exhibit similar behavioral characteristics to previous studies. In addition, some new aspects of the behavior could be systematically tested in HABITS due to its key advantage of autonomy. This high level of versatility, combined with the ability to support arbitrary paradigm designs, suggests that more specialized behavioral paradigms could potentially benefit from HABITS to enhance experimental novelty.

## Innovating mouse behaviors in HABITS

One of the main goals of HABITS was to expand mouse behavioral reservoir by developing complex and innovative paradigms that had previously proven challenging or even impossible for mice. These paradigms imposed higher cognitive abilities demands, which required an extensively long period to

test in a mouse model. HABITS enabled unsupervised, efficient, and standardized training of these challenging paradigms at scale, and thus was suitable for behavioral innovations.

Firstly, leveraging the autonomy of HABITS, we tested mouse's ability to successively learn up to 5 tasks one after another without any cues (*Figure 4A*). These tasks included 2AFC based on sound frequency, sound frequency reversal, sound orientation (pro), sound orientation (anti), and light orientation (*Figure 4A1*). Firstly, the results showed that mice could quickly switch from one task to another and the learning rates across these sub-tasks roughly followed the learning difficulty of modalities (*Figure 4A2*). Specifically, reversal of sound frequency was cognitively different from reversal of sound orientation (i.e., from pro to anti), which resulted in significantly longer learning duration (*Figure 4A2*). Secondly, mice dealt with new tasks with higher reaction time and gradually decreased as training progressed (*Figure 4A3*). It implied a uniform strategy mice applied: mice chose to respond more slowly in order to learn quickly (*Masís et al., 2023*). Lastly, mice exhibited large bias at the beginning of each task in all tasks, including tasks without reversals (*Figure 4*). This means that mice acquired reward only from one lickport in the early training and switched strategy to follow current stimulus gradually, which implied a changing strategy from exploration to exploitation.

The delayed match sample task was quite challenging for mice, and only olfactory and tactile modalities were implemented previously *Condylis et al., 2020*; *Liu et al., 2014*; *Taxidis et al., 2020*. Recently, auditory modality was introduced but only in a go/no-go paradigm *Yu, 2021*. We next constructed a novel double delayed match sample task (dDMS) task (*Figure 4B*), which required mice to keep working memory of first sound frequency (low or high) during the first delay, match to the second sound based on XOR rules, make a motor planning during the second delay, and finally make a 2AFC choice (*Figure 4B1*). All the 10 mice achieved the performance criteria during the automated training process, though, an averaged 64.45±7.88 days was required, which was equivalent to more than 120,000 trials (*Figure 4B2-3*). After training, the four trial types (i.e. four combinations of frequencies) achieved equally well performance (*Figure 4B4*). During the testing stage, we systematically randomized the duration of the two delays (ranging from 1 to 3 s) and revealed increased error rates and early lick rates as the delay increased (*Figure 4B5*). Challenging tasks that demanded months of training were well suited for HABITS; otherwise, they were difficult or even impossible for manual training.

Subsequently, we attempted to expand the choice repertoire of d2AFC into 3-alternative forced choice (d3AFC), utilizing the three lickports installed in HABITS (*Figure 4C*). Previous studies have implemented multiple choice tasks, but only based on spatial modalities *Asinof and Paine, 2014*; *Birtalan et al., 2020*; *Piantadosi et al., 2019*. In our system, mice learned to discriminate low (8 kHz), medium (16 kHz), and high (32 kHz) sound frequencies and lick respectively left, middle, and right lickport to get reward (*Figure 4C1*). Mice needed to construct two psychological thresholds to conduct correct choices. We successfully trained all the 14 mice to perform the tasks in 13.85±1.05 days, with similar performance among the left, middle, and right trial types (*Figure 4C2*). The final correct rate and early lick rate made no differences for the three trial types (*Figure 4C3*). Interestingly, mice made error choices more in the most proximity lickport; for the middle trials, mice made error choices equally in the left and right sides (*Figure 4C4*). In addition, we tested the whole spectrum of sound frequencies between 8 and 32 kHz and found that mice presented two evident psychological thresholds to deal with this three-choice task (*Figure 4C5*). Finally, we also implemented sound orientation-based d3AFC in another separated group of mice, which actually required longer training duration (45.16±11.46 days) (*Figure 3—figure supplement 1B*). The d3AFC was also tested for reversal contingency paradigm, and an accelerated learning was revealed (*Figure 3—figure supplement 1C*), which potentially provides new insight into the cognitive flexibility *Piantadosi et al., 2019*.

Finally, we introduced one of the most challenging cognitive tasks in mouse model, delayed context-dependent attention task, in HABITS (*Figure 4D*). This task was previously implemented by light and sound orientation modalities *Mukherjee et al., 2021*; *Wimmer et al., 2015*; however, due to the difficulty, it was not well repeated broadly. In HABITS, we tailored this task into a sound-only-based but multimodal decision-making task (*Figure 4D1*). We constructed this task using three auditory modalities: regular clicks (16 vs 64 clicks/s) as context, sound frequency (3 k *vs.* 12 kHz), and sound orientation (left vs. right) as the two stimulus modalities. Mice needed to pay attention to one of the modalities, which presented simultaneously during sample epoch, according to the context cue indicated by the clicks (low click rate to sound frequency and high rate to sound orientation), and make

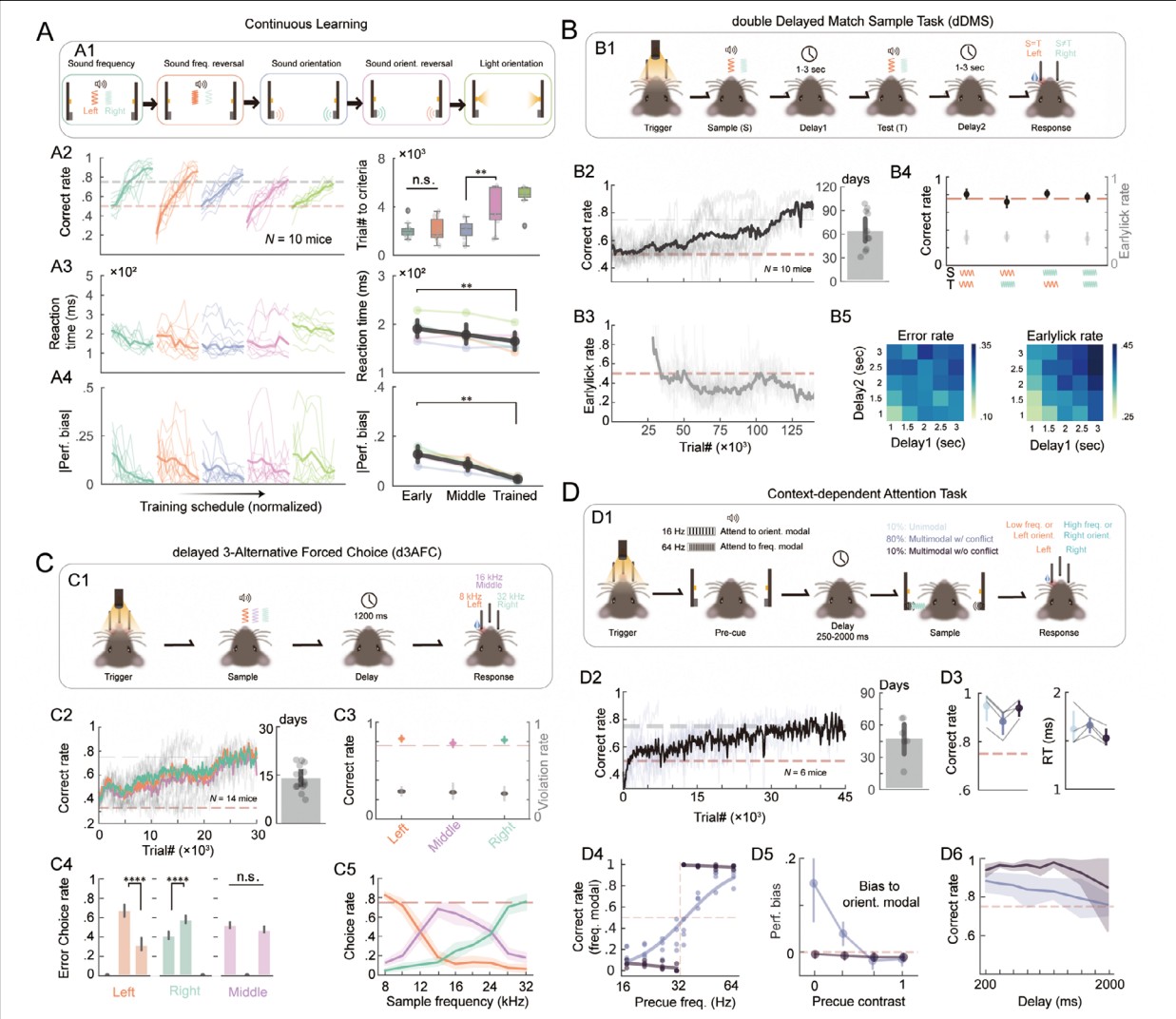

**Figure 4.** Challenging mouse tasks innovated in HABITS. (**A**) Continuous learning task. (**A1**) Task structure showing mice learning five subtasks one by one. (**A2**) Left, averaged correct rate of all mice performing the five tasks (different colors) continually. All task schedules are normalized to their maximum number of trials and divided into 10 stages equally. Right, box plot of number of trials to criteria for each task. (**A3**) Left, averaged reaction time of all mice performing the five tasks continually. Right, averaged median reaction time across the five tasks during early (perf. <0.55), middle (perf. <0.75), and trained (perf. >0.75) stage. Error bar indicates 95% CI. (**A4**) Same as (**A3**) but for absolute performance bias. n.s., p>0.05; **, p<0.01, two-sided Wilcoxon signed-rank tests. (**B**) Double delayed match sample task (dDMS) with sound frequency modality. (**B1**) Task structure. (**B2**) Averaged correct rate across all mice during training (left) and averaged number of days to reach the criterion (right). (**B3**) Averaged early lick rate across all mice. (**B4**) Averaged correct rate (black) and early lick rate (gray) for all combination of sample and test stimulus. (**B5**) Heatmap of error rate (left) and early lick rate (right) varies with different combination of delay1 and delay2 durations. (**C**) Delayed 3 alternative forced choice (d3AFC). (**C1**) Task structure. (**C2**) Averaged correct rate across all mice during training (left, colors indicate trial types) and averaged number of days to reach the criterion performance (right). (**C3**) Averaged correct rate (colors indicate trial types) and early lick rate (gray) for different trial types. (**C4**) Averaged error rate of choices conditioning trial types. In each subplot, the position of bars corresponds to different choices. ****, p<0.0001, n.s., p>0.05, two-sided t-tests. (**C5**) Averaged choice rates for the three lickports (colors) as a function of sample frequency. Data collected from trained mice. (**D**) Context-dependent attention task. (**D1**) Task structure. (**D2**) Averaged correct rate across all mice during training (left, data only from trials with multimodal w/ conflict) and averaged number of days to reach the criterion (right). (**D3**) Correct rate (left) and reaction time (right) conditioning modalities. (**D4**) Averaged psychometric curve and partitioned linear regression for the multimodal with and without conflict conditions, respectively. (**D5**) Performance bias to sound orientation modal as a function of pre-cue contrast, for the two multimodal conditions. (**D6**) Averaged correct rate as a function of delay duration.

a 2AFC decision accordingly. We successfully trained all six mice enrolled in this task, with an average of 48.09±7.54 days (*Figure 4D2*). To validate the paradigm's stability and effectiveness, the direction of stimulus features was presented randomly and independently during the final testing stage. Mice exceeded criterion performance across different trial types (i.e. unimodal, multimodal w/ conflict,

multimodal w/o conflict), indicating effective attention to both stimulus features (*Figure 4D3*). Trials with conflicting stimulus features, requiring mice to integrate context information for correct choice, exhibited reduced decision speeds and accuracy compared to trials without conflicting for all tested mice (*Figure 4D3*). We further systematically varied the click rate from 16 to 64 Hz to change context contrast. For trials with conflicts, mice decreased their accuracy following the decline of context contrast, formulating a flat psychometric curve. However, for trials without conflicts, mice performed as a near-optimal learner (*Figure 4D4*). Meanwhile, as the contrast decreased, mice tended to bias to orientation feature against frequency in conflicting trials, but not for the trials without conflicts (*Figure 4D5*). All these results represented mice dealt with different conditioned trials by a dynamic decision strategy synthesizing context-dependent, multimodal integration, and perceptual bias. Lastly, the performance of both trial types declined with increased delay duration but maintained criterion above up to 2 s delay (*Figure 4D6*), confirming mice could execute this paradigm robustly in our system.

As a summary, by introducing changes in trial structure, cognition demands, and perceptual modalities, we extended mice behavior patterns in HABITS. These behaviors were usually challenging and very difficult to test previously with manual training. Thus, the training workflow of our system potentially allows for large-scale and efficient validation and iteration of innovative paradigms aimed to explore unanswered cognitive questions with mouse models.

## Machine-learning-aided behavioral optimization in HABITS

Mice can be trained to learn challenging tasks in a fully autonomous way in HABITS; however, whether the training efficiency is optimal was unknown. We hypothesized that an optimal train sequence generated by integrating all histories could enhance training procedure, compared with commonly used random or anti-bias strategies. Benefited from recent advances in machine teaching (MT) *Zhu et al., 2018*, and inspired by previous simulations in optimal animal behavior training experiments *Bak et al., 2016*, we developed a MT-based automated training sequence generation framework in HABITS to further improve the training qualities.

*Figure 5A* illustrated the architecture of the MT-based training framework. Initially, mice made an action corresponding to the stimuli presented in current trial *t* in HABITS; subsequently, an online logistic regression model was constructed to fit the mice's history choices by weighted sum of multiple features including current stimulus, history, and rules. This model was deemed as the surrogate of the mouse and was used in the following steps; finally, sampling was performed across the entire trial type repertoire and the fitted model predicted positions of potential future trials in the latent weight space; the trial type with closest position to the goal was selected as the next trials. This entire process forms a closed-loop behavioral training framework, ensuring that the mice's training direction continually progresses towards the goal.

We first validated the theoretical feasibility and efficiency of the algorithms in simulated 2AFC experiments (*Figure 5—figure supplement 1*). Faster increasement in sensitivity to current stimuli was observed through effective suppression of noise-like biases and history dependence with the MT algorithm (*Figure 5—figure supplement 1A–B*). Notably, if the learner was ideal (i.e. without any noise), there was no difference between random and MT strategies to train (*Figure 5—figure supplement 1C*). This implied that the training efficiency was improved by suppression of noise in MT.

To demonstrate MT in real animal training, we initially tested a working memory task similar to *Figure 3B*, but with a fully stimulus generation matrix. As shown in *Figure 5B*, this task utilized a complete set of twenty trial types (colored dots), categorized into four levels of difficulty (dot sizes) according to the distance from decision boundary. Trial type selection using a MT algorithm against a baseline of random selection was tested in two separate groups. Mice trained with the MT algorithm achieved criteria performance with significantly fewer trials compared with the random group (*Figure 5C*); three out of the eight mice in the random group even did not reach the criteria performance at the end of training (60 days). We then asked what kind of strategy the algorithm used that supported an accelerated learning. Analysis of the trial type across learning revealed that the MT-based training presented easier trials first, then gradually increased the difficulty, that is exhibiting a clear curriculum learning trajectory (*Figure 5D*). However, this did not mean that the MT only presented easy trials at the beginning; hard trials were occasionally selected when the model deemed that a hard trial could facilitate the learning. This strategy enabled mice to maintain consistently

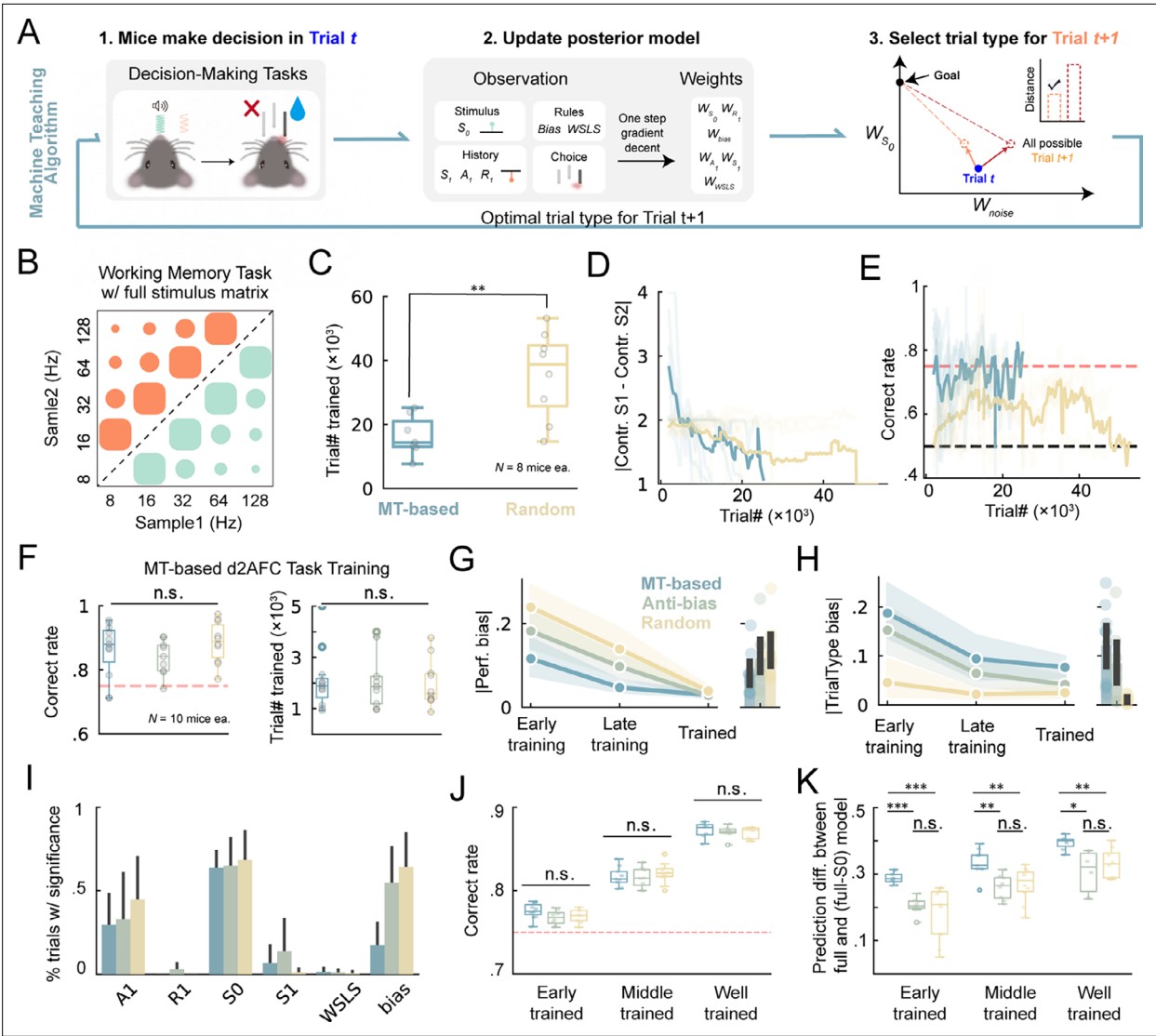

**Figure 5.** MT enabled faster learning with higher quality. (**A**) The framework of machine teaching (MT) algorithm (see text for details). (**B**) Working memory task as in *Figure 4A*, but with full stimulus generation matrix. (**C**) Averaged number of trials needed to reach the criterion for MT-based and random trial type selection strategies. **, p<0.01, two-sided Wilcoxon rank-sum test. (**D**) The absolute difference between contrast (contr.) of sample1 (**S1**) and sample2 (**S2**) during training process for the two strategies. (**E**) Same as (**D**) but for correct rate. (**F**) MT-based d2AFC task training. Box plot of correct rate of expert mice (left) and number of trials needed to reach the criterion (right) for different training strategies (MT, anti-bias, and random). n.s., p>0.05, Kruskal–Wallis tests. (**G**) Left, averaged absolute performance bias for the three strategies during different training stages. Right, averaged across training stages. (**H**) Same as (**G**) but for absolute trial type bias. (**I**) Percentage of trials showing significance for different regressors during task learning. (**J–K**) Box plot of correct rate (**J**) and prediction performance difference between the full model and partial model excluding current stimulus (**S0**) (**K**) for different trained stage, including early (perf. >75%), middle (perf. >80%), and well (perf. >85%) trained. *, p<0.05, **, p<0.01, ***, p<0.001, n.s., p>0.05, two-sided Wilcoxon rank-sum tests with Bonferroni correction.

The online version of this article includes the following figure supplement(s) for figure 5:

**Figure supplement 1.** Simulation of machine teaching algorithm in decision-making scenario.

higher performance than random groups throughout the training process (*Figure 5E*). These results suggested that MT-based method enabled more efficient training for specific challenging tasks.

To further validate the effectiveness of MT in more generalized perceptual decision-making tasks, we trained three groups of mice using random, antibias, and MT strategy, respectively, in sound-frequency-based 2AFC task. Due to the fact that this task was relatively simple, all three groups of mice achieved successful training, with comparable efficiency and final performance (*Figure 5F*). But interestingly, the MT algorithm effectively reduced mice's preference towards a specific lickport

(i.e. bias; *Figure 5G*) throughout the training process by generating trial types with opposite bias more aggressively (*Figure 5H*). Using a model-based methodology, we demonstrated that while the MT algorithm minimized bias dependency, it did not increase, and even decreased, mice's reliance on other noise variables, like previous action $A_1$, reward $R_1$ and stimulus $S_1$ (*Figure 5I*). Notably, we noticed that all trained mice demonstrated similar low bias (*Figure 5G*), while only the MT algorithm still exhibited relatively high anti-bias strategy during the trained stage (*Figure 5H*). This suggests that the MT algorithm might keep regulating cognitive processes actively even in expert mice. To verify this, we segmented the trained trials into early, middle, and well-trained stages based on performance level and showed that all three groups of mice had similar overall accuracies across stages (*Figure 5J*). However, when we examined the reliance on the current stimulus $S_0$, that is to what extent the decision was made according to current stimulus, we found that the MT group had significantly higher weights for $S_0$ than both anti-bias and random groups (*Figure 5K*). This means that MT-generated sequences across all stages encouraged mice to rely only on current stimuli, rather than noise factors. These results suggested that MT-based training had higher quality for both training and trained stage.

In summary, the MT algorithm automatically chose optimal trial type sequences to enable faster and more efficient training. By modeling the effect of history, choice, and other noise behavioral variables, the MT method manifested higher quality training results. Based on these characteristics, the MT algorithm could enhance training efficiency in challenging paradigms and promote testing robustness in more general paradigms.

## Behavioral optimization for multi-dimensional tasks

One of the advantages of the MT algorithm remains that it considers multi-dimensional features altogether and gives the optimal trial sequences to guide the subject to learn. To test this feature, we next expanded the 2AFC task to two stimulus dimensions and trained mice to learn a dynamic

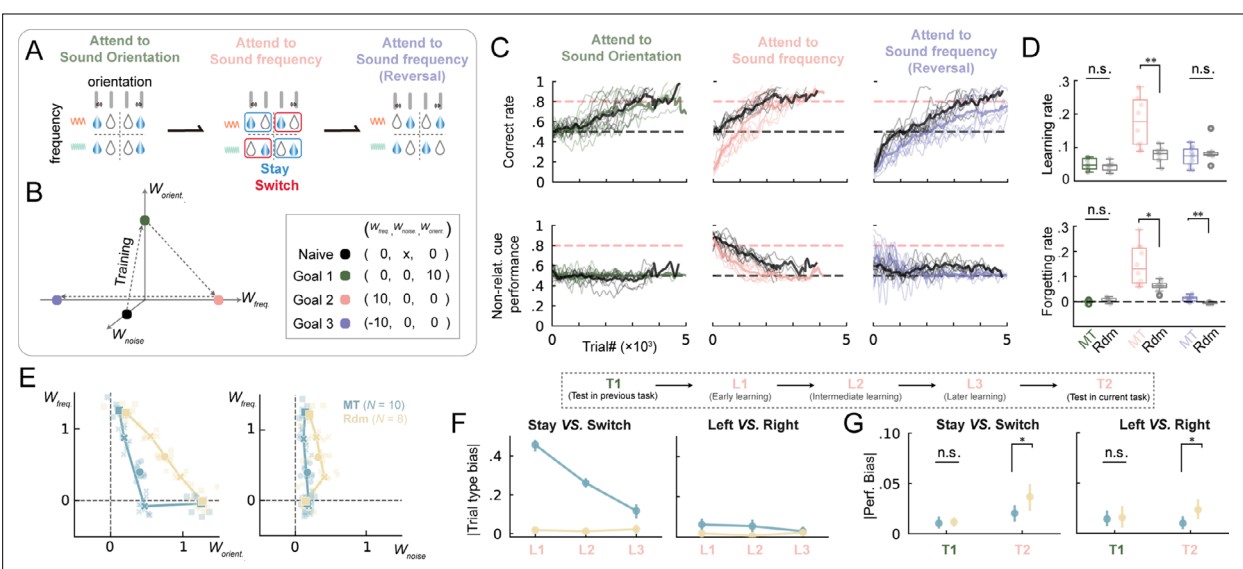

**Figure 6.** MT manifested distinct learning path with faster forgetting and higher learning rate. (**A**) Task structure. (**B**) Chart of training path in latent decision space following three goals one by one. (**C**) Top, averaged correct rate across grouped mice during training (color, machine teaching; black, random). Bottom, same as top but performance for non-relative cue. (**D**) Top, the slopes of linear regression between trial number and correct rate. Bottom, same as top but between trial number and performance for non-relative cue. **, p<0.01; n.s., p>0.05; two-sided Wilcoxon rank-sum tests. (**E**) The learning path of mice (lines) in latent decision space for machine teaching and random training strategies. Light dots represent model weights fitted by individual mice's behavioral data. Shaded dots, averaged across mice. (Square dots, testing protocol; Cross dots, the first or the last half of trials in learning protocol; Cycle dots, all trials in learning protocol) (**F**) Left, averaged absolute trial type bias between stay and switch conditions across grouped mice for the MT and random strategies from L1 to L3. Right, same as middle but for the bias between left and right trials. (**G**) Same as (**H**) but for absolute performance bias in T1 and T2 protocols. L1, the first 500 trials of frequency learning protocol; L2, intermediate trials of frequency learning protocol; L3, the last 200 trials of frequency learning protocol; T1, testing orientation protocol; T2, testing frequency protocol. *, p<0.05; n.s., p>0.05; two-sided t-tests.

The online version of this article includes the following figure supplement(s) for figure 6:

**Figure supplement 1.** Details of behavioral analysis for multi-dimensional tasks.

stimuli-action contingency. The task was similar to the one applied in *Figure 4D* which presented both sound frequency and orientation features but without context cues. As illustrated in *Figure 6A*, the mice were initially required to focus on sound orientation to obtain reward, while ignoring the frequency (Goal 1). Subsequently, the stimulus-action contingency changed, making sound frequency, rather than orientation, the relevant stimuli dimension for receiving reward (Goal 2). Finally, the relevant cue was still sound frequency but with reversed stimulus-action contingency (Goal 3). Throughout the training process, we employed the MT algorithm to adaptively generate trial types about not only the reward location (left or right) but also the components of the sound stimuli (frequency and orientation combinations) and compared with the random control group. The MT algorithm allowed for the straightforward construction of a dynamic multi-goal training just by setting the coordinates of target goals within the latent weight space (*Figure 6B*).

Both groups of mice successfully completed all the goals, achieving an overall correct rate of over 80% (*Figure 6C*). During the first and the third goal of training, the learning rates of mice in both groups were similar, showing low sensitivity to irrelevant cue. However, in the second goal of training (i.e. transition from orientation to frequency modality), the MT group exhibited a significantly higher learning rate compared to the random group, along with a significantly faster rate of forgetting of the irrelevant cue (*Figure 6D* and *Figure 6—figure supplement 1A–D*). To construct a paired comparison, we also retrained the MT group mice in the same protocols but with random sequences after forgetting and confirmed the improvements for both the learning rate and training efficiency (*Figure 6—figure supplement 1F–G*). We again employed the logistic regression model to extract the weights for each variable and plotted the learning trajectories in the latent weight space. MT algorithm manifested distinct learning path against random group in the space (*Figure 6E*); MT algorithm quickly suppressed the sensitivity of irrelevant modality (i.e. $W_{orient.}$), keeping low sensitivity to noise dimensions (i.e. $W_{noise}$) in the meantime. This resulted in a circuitous learning path in the space compared to the random group.

We then asked how MT achieved this learning strategy and what the benefit is. Compared to the random sequence, the MT algorithm effectively suppressed the mice's reliance on irrelevant strategies by dynamically adjusting the ratio of stay/switch trials and left/right trial types (*Figure 6F* and *Figure 6—figure supplement 1E*). After training, we employed the same random trial sequence to test the performance of both groups. Notably, those mice trained with the MT algorithm exhibited significantly lower left/right bias and stay/switch bias compared to the randomly trained mice (*Figure 6G* and *Figure 6—figure supplement 1H*). This suggested that the MT algorithm enabled mice to exhibit more stable and task-aligned behavioral training, which implied an internal influence to psychological decision strategies after the MT conditioning.

## Discussion

In this study, we developed a fully autonomous behavioral training system known as HABITS, which facilitates free-moving mice to engage in 24/7 self-triggered training in their home-cage without the need for water restriction. The HABITS is equipped with a versatile hardware and software framework, empowering us to swiftly deploy a spectrum of cognitive functions, including motor planning, working memory, confidence, attention, evidence accumulation, multimodal integration, etc. Leveraging the advantages of long-term and parallel running of HABITS, we explored several challenging and novel tasks, some of which introduced new modalities or had never been previously attempted in mouse models. Notably, the acquisition of several tasks spanned more than three months to learn. Benefited from the power of machine teaching algorithms, we endowed the automated training in HABITS with an optimal training sequence which further significantly improved the training efficiency and testing robustness. Furthermore, we extended the machine teaching algorithm to a more generalized form, incorporating multi-dimensional stimuli, which has resulted in diverse training trajectories in the latent space and thus elevated training qualities. Altogether, this study presents a fully autonomous platform that offers innovative and optimal mouse behaviors that could advance the field of behavioral, cognitive, and computational neuroscience.

One of the pivotal contributions of this research is the provision of an extensive behavioral dataset, derived from ~300 mice training and testing in more than 20 diverse paradigms. This comprehensive dataset consisted of the entire learning trajectory with well-documented training strategies and behavioral outcomes. This unprecedented scale of data generated in a single study is mainly attributed to

the three distinct features of the HABITS system. Firstly, the hardware was engineered to fit a broad spectrum of cognitive tasks for mice, diverging from the typical focus on specific tasks in previous studies (*Aoki et al., 2017*; *Bernhard et al., 2020*; *Bollu et al., 2019*; *Caglayan et al., 2021*; *Francis et al., 2019*; *Francis and Kanold, 2017*; *Hao et al., 2021*; *Kiryk et al., 2020*; *Murphy et al., 2020*; *Murphy et al., 2016*; *Poddar et al., 2013*; *Qiao, 2019Qiao, 2019*; *Salameh et al., 2020*; *Silasi et al., 2018*). It integrated both visual and auditory stimuli across three spatial locations, as well as up to three lickports for behavioral report, offering various combinations to explore. Secondly, the software within HABITS implemented a general-purpose state machine runner for step-by-step universal task learning and ran standalone without PC in the loop, which contrasted with previous system running on PCs (*Aoki et al., 2017*; *Caglayan et al., 2021*; *Francis et al., 2019*; *Francis and Kanold, 2017*; *Poddar et al., 2013*). Thirdly, the cost of the HABITS was quite low (less than $100) compared to previous systems (ranging from $250 to $1500; *Francis et al., 2019*; *Francis and Kanold, 2017*; *Qiao, 2019Qiao, 2019*; *Silasi et al., 2018*), which facilitated large-scale deployment and thus high-throughput training and testing. Together, these unique attributes have simplified behavioral exploration, which would otherwise be a time- and labor-intensive endeavor.

Another significant contribution was the expansion of the behavioral repertoire for mice, made possible again by the autonomy of HABITS. We have introduced auditory stimuli and multiple delay epochs into the DMS paradigm (*Condylis et al., 2020*; *Liu et al., 2014*; *Taxidis et al., 2020*; *Yu, 2021Yu, 2021*), allowing for investigation of working memory across different modality and stages within single trials. Additionally, we have advanced the traditional d2AFC task (*Guo et al., 2014b*; *Inagaki et al., 2018*) to d3AFC, enabling the study of motor planning in multi-classification scenarios. Furthermore, we have implemented delayed context-dependent tasks *Mukherjee et al., 2021*; *Wimmer et al., 2015* based on multi-dimensional auditory stimuli, facilitating research of complex and flexible decision-making processes. Collectively, the advancement of our high-throughput platform was anticipated to improve the experimental efficiency and reproducibility, in either the creation of standardized behavioral datasets for individual paradigms or in the exploration of a multitude of complex behavioral training paradigms.

Last but not least, we have incorporated machine teaching algorithms into the mouse behavioral training process and significantly enhanced the training efficacy and quality. To our knowledge, this is the first study demonstrating the utility of machine teaching in augmenting animal behavioral training, which complements previous simulation studies *Bak et al., 2016*. The impact of machine teaching algorithms is threefold. First, the training duration of complex tasks was substantially reduced, primarily due to the real-time optimization of trial sequences based on mouse performance, which significantly reduces the mice's reliance on suboptimal strategies. Second, the final training outcomes were demonstrated to be less influenced by the task-irrelevant variables. A prior study has indicated that suboptimal strategies, such as biases, are common among expert mice trained in various paradigms, potentially stemming from their exploration in real-world uncertain environments (*Pisupati et al., 2021*; *Rosenberg et al., 2021*; *Wang et al., 2023*; *Zhu and Kuchibhotla, 2024*). Machine teaching-based techniques can significantly reduce the noise dependency, thus facilitating the analysis of the relationship between behavior and neural signals. Third, the machine learning algorithm lowers the barriers for designing effective anti-bias strategies, which were challenging and prone to lopsided in multidimensional tasks. By simply setting the task goals in the fitted decision model, machine teaching can automatically guide the mouse to approach the goal optimally and robustly.

Our study was designed to standardize behavior for the precise interrogation of neural mechanisms, specifically addressing within-subject questions. However, investigators are often interested in between-subject differences—such as sex differences or genetic variants—which can have long-term behavioral and cognitive implications (*Lassalle et al., 2008*; *Weekes, 1994*). This is particularly relevant in mouse models due to their genetic tractability (*Hemann et al., 2012*). Although our primary focus was not on between-subject differences, the dataset we generated provides preliminary evidence for such investigations. Several behavioral readouts revealed individual variability among mice, including large disparities in learning rates across individuals (*Figure 2I*), differences in overall learning rates between male and female subjects (*Figure 2D* vs. *Figure 2—figure supplement 1G*), variations in nocturnal behavioral patterns (*Figure 2K*), etc. Furthermore, a detailed logistic regression analysis dissected the strategies mice employed during training (*Figure 2—figure supplement 1B*). Notably, the regression identified variables associated with individual task-performance strategies

(*Figure 2—figure supplement 1C*), which also differed between manually and autonomously trained groups (*Figure 2—figure supplement 1D*). Thus, our system could facilitate high-throughput behavioral studies exploring between-subject differences in the future.

Our study marks the inaugural endeavor to innovate mouse behavior through autonomous setups, yet it comes with several limitations. Firstly, our experiments were confined to single-housed mice, which is known to influence murine behavior and physiology, potentially affecting social interaction and stress levels (*Arndt et al., 2009*). In our study, individual housing was necessary to ensure precise behavioral tracking, eliminate competitive interactions during task performance, and maintain consistent training schedules without disruptions from cage-mate disturbances. However, the potential of group-housed training has been explored with technologies such as RFID *Kiryk et al., 2020*; *Murphy et al., 2020*; *Qiao, 2019*; *Salameh et al., 2020*; *Silasi et al., 2018* to distinguish individual mice, which potentially improves the training efficiency and facilitates research of social behaviors (*Torquet et al., 2018*). Notably, it has shown that simultaneous training of group-housed mice, without individual differentiation, can still achieve criterion performance (*Francis et al., 2019*). Secondly, we have not yet analyzed any videos or neural signals from mice trained in the home-cage environment. Recent studies have harnessed a variety of technologies and methodologies to gain a deeper understanding of natural animal behavior in home-cage environments (*Grieco et al., 2021*; *Jhuang et al., 2010*). Voluntary head fixation, employed in previous studies, has facilitated real-time brain imaging (*Rich et al., 2024*; *Scott et al., 2013*; *Aoki et al., 2017*; *Murphy et al., 2020*; *Murphy et al., 2016*). Future integration of commonly used tethered, wireless head-mounted (*Shin et al., 2022*), or fully implantable devices (*Ouyang et al., 2023*; *Wright et al., 2022*), could allow for investigation of neural activity during the whole period in home-cage. Lastly, while HABITS achieves criterion performance in a similar or even shorter overall days compared to manual training, it requires more trials to reach the same learning criterion (*Figure 2G*). We hypothesize that this difference in trial efficiency may stem from the constrained engagement duration imposed by the experimenter in manual training, which could compel mice to focus more intensely on task execution, resulting in less trial omissions (*Figure 2F*). In contrast, the self-paced nature of autonomous training may permit greater variability in attentional engagement (*Smolen et al., 2016*) and inter-trial intervals, which could be problematic for data analysis relying on consistent intervals and/or engagements. Future studies should explore how controlled contextual constraints enhance learning efficiency and whether incorporating such measures into HABITS could optimize its performance.

The large-scale autonomous training system we proposed can be readily integrated into current fundamental neuroscience research, offering novel behavioral paradigms, extensive datasets on mouse behavior and learning, and a large cohort of mice trained on specific tasks for further neural analysis. Additionally, our research provides a potential platform for testing computational models of cognitive learning, contributing to the field of computational neuroscience.

## Materials and methods
### Design and implementation of HABITS
#### Architecture

A single HABITS was comprised of a custom home-cage and integrated behavioral apparatuses. All the building materials were listed in the *Supplementary file 1*, with source and price information provided. The home-cage was made of acrylic panels, with a dimension of 20×20 × 30 cm³. The top panel was movable and could be equipped with cameras to record mouse natural behaviors. A compatible tray was located at the bottom of the home-cage, facilitating bedding materials changing. A notch was designed in the front of the tray where an elevated platform was installed. The platform formed an arch shape to loosely constrain the mouse body when the mouse stepped on it to perform the task. A micro load cell was installed beneath the platform and used for daily body weighing.

Most of the behavioral apparatuses were installed in the front panel of the home-cage. A lickport holder with up to seven slots was installed in front of the weighting platform. Three lickports (1.5 mm diameter, 10 mm apart) were used in this study. Water was drawn by peristaltic pumps from water tanks (centrifuge tube, 50 ml) to the lickports. Three groups of LEDs and buzzers for light and sound stimuli were extruded from the front panel and placed in the left, right, and top positions around the weighting platform. Notably, the top module contained an RGB LED, but white LEDs for the others.

Buzzers were the same in all stimulus modules and produced 3–15 kHz pure tones at 80 dB. In some experiments (*Figures 3E and 4C*), the top buzzer was replaced with a micro ultrasound speaker (Elecfans Inc) which was able to emit 40 khz pure tone for up to 100 dB.

## Control system

The core of the control system was a microprocessor (Teensy 3.6) which interacted with all peripheral devices and coordinated the training processes (*Figure 1—figure supplement 1A*). The microprocessor generated a PWM signal to directly control the sound and light stimuli. Reward water was dispersed by sending pulses to solid-state relays which controlled the pumps. Two toggle switches were used for flushing the tubing. Each lickports was electrically connected to a capacitive sensing circuit for lick detection. Additionally, another switch was used for manually controlling the start and pause of the training process. Real-time weight data were read from the load cell at a sampling rate of 1 Hz. A Wi-Fi module was connected with the microprocessor to transmit data wirelessly to a host computer. Meanwhile, all the data were also stored on a local SD card, with the microprocessor's clock as the timestamps for all behavioral events.

We have developed a software framework for constructing behavioral training programs, which is a general-purpose state machine runner for training animal behaviors (gpSMART, https://github.com/Yaoyao-Hao/gpSMART, copy archived at *Hao, 2024b*). This framework supported the construction of arbitrarily complex cognitive behavioral paradigms as state machines (*Figure 1—figure supplement 1B*). Basically, each state was comprised of a unique name, output actions, transition conditions, and maximum timing. Within each trial, the microprocessor generates the state matrix based on the defined state machine and executes the state transition according to the external events (e.g. licks) or timing (e.g. a delay period of 1.2 s). This is similar to the commonly used Bpod system (Sanworks Inc), but gpSMART could run on microprocessors with a hardware-level time resolution. Between trials, training protocol updating, behavioral data recording, and wireless data communication were executed. Various training assistances (e.g. free reward) were also performed when necessary to help the training processes. All the training progress and protocols were stored on the SD card for each mouse; thus, training can be resumed after any pause event and supports seamless switching between multiple HABITS systems. The system was designed to operate standalone without a PC connected. Finally, the firmware on the microprocessor could be updated wirelessly to facilitate paradigm changing.

## High-throughput training and GUI

We constructed over a hundred of HABITS to facilitate large-scale, fully autonomous in-cage behavioral training (*Figure 1—figure supplement 1D*). Each HABITS was piled on standard mouse cage racks, with sound-proof foams installed between them to minimize cross-cage auditory interference. The cage operated independently with each other, with only a 12 V standard power supply connected. The training room was maintained under a standard 12:12 light-dark cycle. All the cages communicated with a single PC via unique IP addresses using the UDP protocol.

To monitor the training process of all the cages, a MATLAB-based GUI running on a PC was developed (*Figure 1—figure supplement 1C*). The GUI displayed essential information for each mouse, such as the paradigm name, training duration, training progress, and performance metrics like long-term task accuracy, weight changes, and daily trial numbers, etc. Meanwhile, the whole history of training performance, detailed trial outcomes in the last 24 hr and real-time body weight could be plotted. The GUI also enabled real-time updating of each cage's training parameters for occasional manual adjusting. Training settings can also be modified by physically or remotely updating the SD card files.

## Mice and overall training procedures

### Mice

All experimental data used in this study were collected from a total of 302 mice (C57BL/6 J). For most of the autonomous experiments, males were used with starting age at around 8 weeks (see *Table 1*). A separate group of six females was tested in a sound-frequency-based 2AFC task (*Figure 2—figure supplement 1G*). Mice were single housed in our home-cage systems for ranging from 1 to more than 3 months. A group of six mice was used for supervised manual training. Another six mice were used

for ad libitum reward testing in HABITS. All experiments were approved by the Laboratory Animal Welfare and Ethics Committee of Zhejiang University (Ethics Code: ZJU20210298).

## Workflow for behavioral testing in HABITS

The entire workflow for fully automated behavioral training experiment in HABITS can be divided into three stages (*Figure 1—figure supplement 1E*). The first stage was the initialization of HABITS. This involved setting up the home cage by placing an appropriate amount of bedding, food, cotton, and enrichments into the drawer of the home cage. Behavioral paradigms and training protocols, programmed within our software framework, were then deployed on the microcontroller of HABITS. Each mouse was provided with a unique SD card that stores its specific behavioral training data, including the training paradigm, cage number, initial paradigm parameters, and progresses. The load cell was initialized through the host computer, which includes zeroing and calibration processes. The flush switch of the peristaltic pump was activated to fill the tubing with water. Finally, the mouse was placed into the HABITS after initial body weight measuring. Note that any habituations or water restrictions were not required.

The second stage was the fully autonomous training phase, during which no intervention from the experimenter was needed. Typically, this stage included three main training sub-protocols: habituation, training, and testing. During the habituation phase, free rewards are randomly given on either lickports to guide the mouse in establishing a connection between lickports and water reward. Subsequently, in the training phase, the protocols are gradually advanced, from very easy ones to the final paradigm, based on the learning performance of the mouse. Assistance, like reward at correct lickport, was gradually decreased as the mouse learned the task. Finally, predefined behavioral tests, such as psychometric curve testing, random trials, additional delays, etc. were conducted. The entire training process of all cages was remotely monitored via the GUI. The bedding in the drawer was replaced every other week to ensure that the mouse lives in a clean environment.

The third stage involved data collection and analysis. All raw data, including detailed event, trail, and configuration information, was stored on the SD card; data wirelessly transmitted to PC were mainly used for monitoring. These behavioral data were analyzed offline with Python, and the mice were ready for other subsequent testing.

## Manual training

To compare with fully autonomous training, we also used HABITS as a behavioral chamber to perform manual training protocol for freely moving mice. The mice were first single-housed in standard home cages and subjected to water restriction. After several days, when the mice's body weight dropped to approximately 85% of their original weight (*Guo et al., 2014a*), behavioral training began. The mice were trained in a session-based manner; in each session, experimenters transferred the mice from the standard home-cage to HABITS, where they underwent 1–3 hr of training to receive around 1 ml of water. The amount of water consumed was estimated by HABITS based on the number of rewards. HABITS weighed the mice daily, ensuring that all mice maintained stable body weight throughout the training process. The manually trained mice underwent the same training protocols as in the autonomous ones (*Figure 2—figure supplement 1A*). Once the mice completed the final protocol and reached the criterion performance (75%), they were considered successfully trained. After completing the manual training, the mice were then transitioned into autonomous testing in HABITS (*Figure 2—figure supplement 1E*).

## Behavioral data analysis
### Bias calculation

We calculated mice's bias toward different trial types, for example left and right, by evaluating their performance under these trial types (perf. bias). The strength of the bias was quantified by calculating the absolute difference between the proportion of performance under specific trial type relative to the summed value across trial types, and the balance point, that is 50%. Similarly, we applied this method for presentation of trial sequence to compute the trial type bias during paradigm training, illustrating the dynamic changes in training strategies.

## Data preprocessing

For the fully autonomous training, we excluded data from the habituation phase, as we believed the mice had not yet understood the structure of the trials during that stage. Additionally, we removed

trials where the mice did not make a choice, that is, no-response trials. For each mouse, the trials were concatenated in chronological order, ignoring the time span between trials during the continuous multi-day home-cage training sessions; the same approach was applied to manual training data. We then organized the data for each mouse into multiple 500-trial windows, sliding from the beginning to the end of the training with a step size of 100-trial. Windows containing fewer than 500 trials at the end of the dataset were discarded. We assumed that within each window, the mouse employed a consistent strategy, and a new logistic regression model was fit in each window.

## Logistic regression of behavioral data

Similar to our previous study *Hao et al., 2021*, we employed an offline logistic regression model to predict the choices made by the mice (*Figure 2—figure supplement 1B–D*, *Figure 5I*, *Figure 6*). This model calculates a weighted sum of all behavioral variables and transforms the decision variable into a probability value between 0 and 1 using a sigmoid function, representing the probability of choosing the left side. The variables include the current stimulus ($S_0$; –1 for licking left trials; 1 for licking right trials), the previous stimulus ($S_1$), reward ($R_1$; –1 for no reward; 1 for reward), action ($A_1$; –1 for left choice; 1 for right choice), win-stay-loss-switch (*WSLS*; which is $A_1 \times R_1$), and a constant bias term (*bias*). The model can be formulated by the following equation:

$$P(right) = \frac{1}{1 + e^{-(z)}}$$

$$z = \beta_{s_0} S_0 + \beta_{S_1} S_1 + \beta_{R_1} R_1 + \beta_{A_1} A_1 + \beta_{WSLS} WSLS + \beta_{bias}$$

where the $\beta$'s were the weights for the regressors. We used 0.5 as the decision threshold: predictions above 0.5 were classified as right choices, while below were classified as left choices.

Model performance was assessed using 10-fold cross-validation. For each cross-validation iteration, 450 trials were randomly selected as the training set, and gradient descent was employed to minimize the cross-entropy loss function. The remaining 50 trials were used as the test set. Training was considered complete (early stopping) once the calculated loss in the test set stabilized. The accuracy of the model in predicting the mouse's choices in the test set was recorded as the result of one cross-validation iteration. This process was repeated 10 times, and the final performance of the model was averaged across all iterations.

## Significance calculation

To evaluate the contribution of each regressor, we compared the performance of a partial model, where a specific variable was removed, with that of the full model. Specifically, the value of the variable in testing was set to zero, and we checked whether the performance of the partial model showed a significant decline. We applied a corrected t-test using a 10×10 cross-validation model comparison method to compute the *p*-value (*Gardner and Brooks, 2017*). For each window, we trained 100 models, and the performance differences between the partial and full models formed a *t*-distribution. By examining the distribution of performance differences, we determined the significance level of each regression variable's contribution. When p<0.05, the regression variable was considered to have a significant contribution to predicting the mouse's choice in that window. Additionally, we calculated the proportion of windows across the entire training process in which a particular regression variable had a significant contribution, to estimate the degree to which the mouse relied on that variable. The same significance evaluation method was applied to both autonomous and manual training, allowing for direct comparison of the learning strategies employed in two conditions at the individual mouse level (*Figure 2—figure supplement 1B–D*, *Figure 5I*).

## Logistic regression in evidence accumulation: Multimodal integration

For each mouse in evidence accumulation task (*Figure 3 D4*), we trained a group of logistic regression models to estimate the psychophysical kernel. The entire sample period was divided into 25 bins of 40ms each, with each bin assigned a weight to predict the mouse's choice. An event occurring in a bin was set to 1; otherwise, set to 0. We trained 100 pairs of models for each mouse. Each pair of models was trained using 10% total trials randomly. Each pair of models included one model trained on the original data and another trained on data with bin-wised shuffled within each trial. The psychophysical kernel for each mouse was derived by averaging the first 100 models, compared to a baseline kernel

obtained from the second. Finally, we averaged the results across all 13 mice to statistically estimate the temporal dynamics of mice's evidence dependence in this task.

## Behavioral tasks and training protocols

### General training methodology

In the fully autonomous behavioral training process, all mice learn the required behavioral patterns through trials and errors. The training protocols were pre-defined based on experience. Given that the entire training process is long-term and continuous, a free reward is triggered if a mouse fails to obtain water within the last 3- or 6-hr period, ensuring the mouse receives sufficient hydration. Throughout the training process, we employed a custom-designed 'anti-bias' algorithm to avoid mice always licking one side. Basically, we implemented several priority-based constraints to prevent mice from developing a preference for a particular reward direction:

- - *Highest Priority*: If a mouse consecutively made three errors or no response in a specific trial type, the next trial would maintain the same reward direction.
- - *Second Priority*: If three consecutive trials shared the same reward direction (including no-response trials), the reward direction would switch in the subsequent trial.
- - *Third Priority*: The reward direction of the next trial was sampled based on the average performance of left and right trials, using the following formula:

$$P_{\text{left}} = \frac{1}{2} \times \left( \frac{\sum_{i=1}^{n} R_i \, \text{Corr}_i}{\sum_{i=1}^{n} R_i \, \text{Corr}_i + \sum_{i=1}^{n} L_i \, \text{Corr}_i} + \frac{\sum_{i=1}^{n} L_i \, \text{Err}_i}{\sum_{i=1}^{n} L_i \, \text{Err}_i + \sum_{i=1}^{n} R_i \, \text{Err}_i} \right)$$

where $N$ represented the number of recorded historical trials (set to 50 in our case). $Ri$ and $Li$ were set to 1 if the reward direction of the $i_{th}$ historical trial was right or left, respectively; otherwise, they were set to 0. Similarly, $Corr_i$ and $Err_i$ were set to 1 or –1 if the mouse's choice in the $i_{th}$ trial was correct or incorrect, respectively; otherwise, they were set to 0. $P_{left}$ represented the probability of left trial type in the next trial.

## d2AFC with multi-modal

The task of d2AFC, delayed two-alternative forced choice, required mice to learn the stimulus-action contingency separated by a delay for motor planning. A complete trial consisted of three parts: the sample, delay, and response epochs. The sample epoch lasted for 1.2 s and is accompanied by auditory or visual stimuli. The delay epoch elapsed for another 1.2 s, during which mice were required to withhold licking until a 100ms response cue (6 kHz tone) was played. Any premature licking (early licks) during this period immediately paused the trial for 300ms. Response epoch lasted for 1 s. The first lick made by the mouse during this period was recorded as its choice for the current trial, and feedback is provided accordingly. A correct lick delivered approximately 0.25 µl of water to the corresponding spout (achieved by activating the peristaltic pump for 30ms), while an incorrect choice results in an immediate 500ms white noise and 8000ms timeout for penalty. After each trial, the mouse must refrain from licking for 1000ms before the next trial began automatically. If the mouse failed to make a choice during the response period, the trial was marked as a no-response trial, and the mouse must lick either spout to initiate the next trial. The stimuli modalities tested in this study were as follows:

- - Sound frequency modality: A 3 kHz tone corresponds to a left choice, while a 10 kHz tone corresponds to a right choice.
- - Sound orientation modality: A sound from the left speaker corresponds to a left choice, while a sound from the right speaker corresponds to a right choice.
- - Light orientation modality: The left white LED lighting up corresponds to a left choice, and the right white LED lighting up corresponds to a right choice.
- - Light color modality: The top tricolor LED lighting up blue corresponds to a left choice, and red corresponds to a right choice. For light color modality, we tested multiple variations since the mouse did not learn the task very well, including green *vs.* blue and flashed green *vs.* blue.

  In the reaction time version of the paradigm (RT task, *Figure 2—figure supplement 2*), a central spout was introduced in addition to the left and right lickports, to allow the mouse to self-initiate a trial. The mouse must lick the central spout to initiate a trial; licking either side following would result in a brief (100ms) white noise and immediate termination of the

trial, followed by a timeout period. During the sample epoch, a tone from the top speaker (3 kHz for left reward; 10 kHz for right reward) plays for 1000ms. The mouse can immediately indicate its choice by licking either side lickports, which terminated the sample period and triggered trial outcomes as above. An inter-trial interval (ITI) of 1000ms was followed. The next trial required the mouse to lick the central spout again. The central spout did not provide any rewards; all rewards are contingent upon the mouse's choice of the left or right spouts.

## Training and testing of other tasks

The training and testing method for all other tasks was detailed in the text of *Supplementary file 2*.

## Machine teaching algorithms

We employed machine teaching (MT) algorithm *Liu et al., 2017*, to design an optimal trial sequence that enables the mice to rapidly approach a target state, i.e., trained in a specific task. In this context, the MT algorithm can be referred to as the 'teacher' while the mice are the 'students'; the size of the training dataset is termed 'teaching dimension' of the student model *Zhu et al., 2018*. Specifically, the teacher samples from a pre-defined discrete dataset, and the student updates its internal model using the sampled data. The teacher then prepares the next round of training based on the student's progress, creating a closed-loop system. In this study, a logistic regression model was employed to infer the internal decision-making model of the mice based on their choices trial-by-trial (i.e. model-based). The model was updated in real-time and used for the optimization and sampling of subsequent trials. L1 regularization and momentum were introduced to smooth the fitted weights, mitigating overfitting and oscillations. The mice's choices and outcomes served as feedback for MT. *Figure 5A* illustrated the complete closed-loop optimization process of MT algorithm. It was similar to an imitation teacher *Liu et al., 2017* whose objective was to iteratively minimize the distance between the model weights of next trial and target.

In detail, the logistic regression model to fit the choices of mouse was updated trial-by-trial according to the following formula:

$$m^t = (1 - \eta)\left(\langle \omega^t, x^t \rangle - y_{\text{choice}}\right)x^t + \eta m^{t-1} \quad m^0 = 0$$
$$\omega^{t+1} = \omega^t - \alpha\left(\frac{m^t}{1 - \eta^t} + \lambda\,\text{sgn}\left(\omega^t\right)\right)$$

where the parameter $\omega^t$ represents the decision model parameters fitted to the current and past choices (i.e. $y_{choice}$) performed by the mice at the $t$-th trial. The hyperparameter $\lambda$ controls the strength of L1 regularization. Momentum parameter $\eta$ determines the window width for exponential smoothing of the loss gradient. $m$ represents an exponential smoothed gradient across past trials.

Then, the objective of this algorithm can be formalized as the following equation:

$$\left\|\omega^{t+1} - \omega^*\right\|_2^2 = \left\|\omega^t - \omega^*\right\|_2^2 + \gamma^2 T_1\left(x, y \mid \omega^t\right) - 2\gamma T_2\left(x, y \mid \omega^t\right)$$
$$T_1\left(x, y \mid \omega^t\right) = \left\|\frac{1}{1 + e^{\left(y(\omega^t, x)\right)}}\right\|$$
$$T_2\left(x, y \mid \omega^t\right) = \left\langle \omega^t - \omega^*, \left(\langle \omega^t, x \rangle - y\right)x \right\rangle$$

where, $(x, y)$ represents a pair of stimuli-action contingency. The parameter $\omega^*$ denotes the target weight within the implicit decision space of the simulated mouse model, typically set to converge to the model weights according to the current task rules. $T_1$ can be interpreted as the trial difficulty, predicting the probability of incorrect choices performed by mice in this trial, and $T_2$ as the effectiveness of this trial, predicting the correlation between the upcoming mouse behavioral strategy updates and the shortest learning path between $\omega^t$ and $\omega^*$. The balance between these two metrics is achieved through hyperparameters $\gamma$, that is hypothetical learning rate of mouse. In this study, we assume that this model reflects the actual decision-making process of the mice. Subsequently, this algorithm can select the next trial type using the following formula:

$$(x^{t+1}, y^{t+1}) = \underset{x \in X, y \in Y}{\arg\min}(\gamma^2 T_1 - 2\gamma T_2)$$

where *X* and *Y* represent the repertoires of available trial types and action, respectively.

Finally, we presented the selected stimuli *x* in the next trials and expected the mouse to make a correct/incorrect choice *y* and receive a reward/punishment.

We have implemented the aforementioned MT-based optimization algorithm on the microprocessors of HABITS, as a superior alternative to existing 'anti-bias' algorithms. The computation load for running MT optimization on the microprocessors was high, and latency was relatively long compared to trial resolution. However, since the computation was conducted between trials, it did not interfere with the execution of the trials themselves under the gpSMART framework. Additionally, for each mouse, additional files were used to record the hyperparameters and weight changes of the online logistic regression model for each trial. No-response trials were not used for model fitting. However, if the previous trial was a no-response trial, the history-dependence regressors of the current trial were set to 0.

## Simulation experiments

We employed a logistic regression model as the student, tasked with completing a 2AFC task, which involved mapping the current stimulus $S_0$ to a choice while ignoring interference from other features. Another logistic regression model, serving as the imitation teacher, was then used to fit the choices made by the student. Both models operated within the same feature space and utilized the same update algorithm. The hyperparameters were set as follows: $\alpha$=0.1, $\eta$=0.9, $\gamma$=1, and $\lambda$=0.1.

We first simulated the biases and history dependence typically observed in naive mice during the early stages of training by setting the initial values of $S_1$ and *bias* to –2 and 2, respectively. During the trial-by-trial update process, we tracked the changes in the student's weights under the MT algorithm and compared them to those under random training, which served as the baseline. Additionally, we simulated conditions without noise to further examine the differences between the MT algorithm and random training in influencing the student's weight updates.

## Animal experiments 1: Working memory task with full SGM

The first task we tested with the MT algorithm was working memory task with full 5×5 stimulus matrix (*Figure 5B*). The delay duration was set to 500ms. Mice were randomly assigned to two groups: random (N=8) and MT (N=7). Once the mice achieved a 75% correct response rate in the eight most challenging trial types, they advanced to the testing phase. At this stage, only the eight most challenging trial types were randomly presented. When the correct rate reached 75%, the mice were considered to have learned the task. MT used seven features as the inputs for the logistic regression model: *bias*, $S_0$ (the frequency of the first stimulus, where {8, 16, 32, 64, 128} Hz corresponds to values of {−1,–0.5, 0, 0.5, 1}), $T_0$ (the frequency of the second stimulus), $S_1$, $A_1$, $R_1$, and *WSLS*. Hyperparameters were set as follows: $\alpha$=0.01, $\eta$=0, $\gamma$=0.03, and $\lambda$=0.

## Animal experiments 2: 2AFC task

In 2AFC task (*Figure 5F*), mice were divided into three groups, each using different methods to generate the stimulus frequency (3 kHz or 10 kHz) for the next trial: random selection (N=10), the anti-bias strategy (N=10), and the MT algorithm (N=10). The online logistic regression model settings for this task remained consistent with those in *Figure 2—figure supplement 1*. The hyperparameters for the MT algorithm were set as follows: $\alpha$=0.1, $\eta$=0.9, $\gamma$=1, and $\lambda$=0.1.

## Animal experiments 3: dynamic 2AFC task with multi-dimensional stimuli

In the third task, we expanded the stimulus dimension of 2AFC with the combination of sound frequency (high and low) and orientation (left and right), to test MT algorithm (*Figure 6A*). In each trial, the stimulus could be one of the four combinations, but whether the mouse should attend to frequency or orientation modality was not informed. In other words, no modality cues were presented for mice, and mice must rely solely on feedback from past trials to identify what the current modality was. Mice were divided into random (N=8) and MT groups (N=10). In the MT group, both the stimulus combinations and reward direction are determined by the MT algorithm simultaneously. The entire task involved learning three different rules one by one, including sound orientation, sound frequency, and reversed sound frequency. When the mice in each group achieved an accuracy of 80% in the first 500 choice trials under the current rule, they advanced to a testing protocol consisting of at least 100

random trials. When a mouse achieved 80% accuracy in the last 100 trials of the testing protocol, it transitioned to the next rule (*Bernklau et al., 2024*). Hyperparameters were set as follows: $\alpha$=0.1, $\eta$=0.9, $\gamma$=1, and $\lambda$=0.1.

## Statistics

To maximize the utility of HABITS in a wider range of paradigms, we usually employed six mice per paradigm. For experiments where we aimed to conduct between-group comparisons, we increased the sample size to 10 to ensure the stability and reliability of statistical significance. All significance tests were conducted by comparing different groups of animals (e.g. comparing performance levels across different mouse groups). Non-parametric tests, such as the Wilcoxon signed-rank test or rank-sum test, were used for comparisons between two groups, and the Kruskal–Wallis test was used for comparisons among three groups, unless otherwise stated in the figure legends. Data are presented as mean ± 0.95 confidence intervals (CI) across animals, as specified in the figure legends. In the box plots, the center lines represent the median values, the box limits indicate the upper and lower quartiles, the whiskers show 1.5 times the interquartile range, and the points represent outliers. Significance levels are denoted as *, $p<0.05$, **, $p<0.01$, and ***, $p<0.001$ in all figures. All data analyses were performed using Python (version 3.8.13) with the following packages: NumPy (1.21.5), SciPy (1.10.1), matplotlib-inline (0.1.3), pandas (1.3.4), PyTorch (1.11.0), and seaborn (0.13.0).

## Acknowledgements

This work was supported by STI 2030—Major Projects (2021ZD0200405), National Natural Science Foundation of China (62336007), Pioneer R&D Program of Zhejiang (2024C03001), the Starry Night Science Fund of Zhejiang University Shanghai Institute for Advanced Study (SN-ZJU-SIAS-002), and the Fundamental Research Funds for the Central Universities (2023ZFJH01-01, 2024ZFJH01-01).

## Additional information

### Funding

| Funder | Grant reference number | Author |
| --- | --- | --- |
| STI 2030—Major Projects | 2021ZD0200405 | Kedi Xu<br>Yaoyao Hao |
| National Natural Science Foundation of China | 62336007 | Yueming Wang<br>Yaoyao Hao |
| Starry Night Science Fund of Zhejiang University Shanghai Institute for Advanced Study | SN-ZJU-SIAS-002 | Yueming Wang |
| Fundamental Research Funds for the Central Universities | 2023ZFJH01-01 | Yueming Wang |
| Fundamental Research Funds for the Central Universities | 2024ZFJH01-01 | Yueming Wang |
| Pioneer R&D Program of Zhejiang | 2024C03001 | Yaoyao Hao |

The funders had no role in study design, data collection and interpretation, or the decision to submit the work for publication.

### Author contributions

Bowen Yu, Data curation, Software, Formal analysis, Investigation, Visualization, Methodology, Writing – original draft; Penghai Li, Haoze Xu, Data curation, Investigation, Methodology; Yueming Wang, Supervision, Funding acquisition, Project administration; Kedi Xu, Supervision, Funding acquisition, Methodology, Project administration; Yaoyao Hao, Conceptualization, Data curation, Software,

Formal analysis, Supervision, Funding acquisition, Investigation, Visualization, Methodology, Writing – original draft, Project administration, Writing – review and editing

### Author ORCIDs
Bowen Yu ![ORCID] https://orcid.org/0000-0002-1119-0458
Yueming Wang ![ORCID] https://orcid.org/0000-0001-7742-0722
Yaoyao Hao ![ORCID] https://orcid.org/0009-0005-2628-8297

### Ethics
All experiments were approved by the Laboratory Animal Welfare and Ethics Committee of Zhejiang University (Ethics Code: ZJU20210298).

Reviewer #1 (Public review): https://doi.org/10.7554/eLife.104833.3.sa1
Reviewer #2 (Public review): https://doi.org/10.7554/eLife.104833.3.sa2
Reviewer #3 (Public review): https://doi.org/10.7554/eLife.104833.3.sa3
Author response https://doi.org/10.7554/eLife.104833.3.sa4

## Additional files

### Supplementary files
MDAR checklist

Supplementary file 1. List of building materials and prices of HABITS.

Supplementary file 2. The implementation details of tasks tested in HABITS.

### Data availability
The guidance for construction of HABITS, all the training programs, and example behavioral data are available on GitHub (https://github.com/Yaoyao-Hao/HABITS, copy archived at *Hao, 2024a*). The general-purpose state machine runner for training animal behaviors (gpSMART) is also available on GitHub (https://github.com/Yaoyao-Hao/gpSMART, copy archived at *Hao, 2024b*). All data in the main text or the supplementary materials are available on figshare.

The following dataset was generated:

| Author(s) | Year | Dataset title | Dataset URL | Database and Identifier |
| --- | --- | --- | --- | --- |
| Bowen Y, Penghai L, Haoze X, Wang Y, Xu K, Hao Y | 2024 | Behavioral raw data recorded by HABITS autonomously | https://doi.org/10.6084/m9.figshare.27192897.v1 | figshare, 10.6084/m9.figshare.27192897.v1 |

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
