## [Editor Report · eLife Assessment]

This manuscript describes a novel approach for assessing cognitive function in freely moving mice in their home-cage, without human involvement. The authors provide **convincing** evidence in support of the tasks they developed to capture a variety of complex behaviors and demonstrate the utility of a machine learning approach to expedite the acquisition of task demands. This work is **important** given its potential utility for other investigators interested in studying mouse cognition.

---

## [Referee Report · Reviewer #1 (Public review)]

Summary:

This is a new and important system that can efficiently train mice to perform a variety of cognitive tasks in a flexible manner. It is innovative and opens the door to important experiments in the neurobiology of learning and memory.

Strengths:

Strengths include: high n's, a robust system, task flexibility, comparison of manual-like training vs constant training, circadian analysis, comparison of varying cue types, long-term measurement, and machine teaching.

Weaknesses:

I find no major problems with this report.

Comments on revisions:

My concerns have been addressed now.

---

## [Referee Report · Reviewer #2 (Public review)]

Summary:

The manuscript by Yu et al. describes a novel approach for collecting complex and different cognitive phenotypes in individually housed mice in their home cage. The authors report a simple yet elegant design that they developed for assessing a variety of complex and novel behavioral paradigms autonomously in mice.

Strengths:

The data are strong, the arguments are convincing, and I think the manuscript will be highly cited given the complexity of behavioral phenotypes one can collect using this relatively inexpensive ($100/box) and high-throughput procedure (without the need of human interaction). Additionally, the authors include a machine learning algorithm to correct for erroneous strategies that mice develop which is incredibly elegant and important for this approach, as mice will develop odd strategies when given complete freedom.

Weaknesses:

A limitation to this approach is that it requires mice to be individually housed for days to months. This is now adequately addressed in the discussion.

A major issue with continuous self-paced tasks such as the autonomous d2AFC used by the authors is that the inter-trial intervals can vary significantly. Mice may do a few trials, lose interest and disengage from the task for several hours. This is problematic for data analysis that relies on trial duration to be similar between trials (e.g., reinforcement learning algorithms). The authors now provide information regarding task engagement of the mice across a 24 hour cycle (e.g., trials started, trials finished across a 24 h period).

Movies - it would be beneficial for the authors to add commentary to the video (hit, miss trials). It was interesting watching the mice but not clear whether they were doing the task correctly or not. The new videos adequately address these concerns.

The strength of this paper (from my perspective) is the potential utility it has for other investigators trying to get mice to do behavioral tasks. However, not enough information was provided about the construction of the boxes, interface, and code for running the boxes. If the authors are not willing to provide this information through eLife, GitHub, or their own website then my evaluation of impact and significance of this paper would go down significantly. This information is now available to readers.

Minor concerns

Learning rate is confusing for Figure 3 results as it actually refers to trials to reach criterion, and not the actual rate of learning (e.g., slope). This has been modified in the manuscript.

Comments on revisions:

The authors have addressed all my concerns regarding this very exciting manuscript.

---

## [Referee Report · Reviewer #3 (Public review)]

Summary:

In this set of experiments, the authors describe a novel research tool for studying complex cognitive tasks in mice, the HABITS automated training apparatus, and a novel "machine teaching" approach they use to accelerate training by algorithmically providing trials to animals that provide the most information about the current rule state for a given task.

Strengths:

There is much to be celebrated in an inexpensively constructed, replicable training environment that can be used with mice, which have rapidly become the model species of choice for understanding the roles of distinct circuits and genetic factors in cognition. Lingering challenges in developing and testing cognitive tasks in mice remain, however, and these are often chalked up to cognitive limitations in the species. The authors' findings, however, suggest that instead we may need to work creatively to meet mice where they live. In some cases it may be that mice may require durations of training far longer than laboratories are able to invest with manual training (up to over 100k trials, over months of daily testing) but that the tasks are achievable. The "machine teaching" approach further suggests that this duration could be substantially reduced by algorithmically optimizing each trial presented during training to maximize learning.

Weaknesses:

Cognitive training and testing in rodent models fill a number of roles. Sometimes, investigators are interested in within-subjects questions - querying a specific circuit, genetically defined neuron population, or molecule/drug candidate, by interrogating or manipulating its function in a highly trained animal. In this scenario, a cohort of highly trained animals which have been trained via a method that aims to make their behavior as similar as possible is a strength.

However, often investigators are interested in between-subjects questions - querying a source of individual differences that can have long term and/or developmental impacts, such as sex differences or gene variants. This is likely to often be the case in mouse models especially, because of their genetic tractability. In scenarios where investigators have examined cognitive processes between subjects in mice who vary across these sources of individual difference, the process of learning a task has been repeatedly shown to be different. The authors recognize that their approach is currently optimized for testing within-subjects questions, but begin to show how between-subjects questions might be addressed with this system.

The authors have perhaps shown that their main focus is highly-controlled within-subjects questions, as their dataset is almost exclusively made up of several hundred young adult male mice, with the exception of 6 females in a supplemental figure. It is notable that these female mice do appear to learn the two-alternative forced choice task somewhat more rapidly than the males in their cohort, and the authors suggest that future work with this system could be used to uncover strategies that differ across individuals.

Considering the implications for mice modeling relevant genetic variants, it is unclear to what extent the training protocols and especially the algorithmic machine teaching approach would be able to inform investigators about the differences between their groups during training. For investigators examining genetic models, it is unclear whether this extensive training experience would mitigate the ability to observe cognitive differences, or select for the animals best able to overcome them - eliminating the animals of interest. Likewise, the algorithmic approach aims to mitigate features of training such as side biases, but it is worth noting that the strategic uses of side biases in mice, as in primates, can benefit learning, rather than side biases solely being a problem. However, the investigators may be able to highlight variables selected by the algorithm that are associated with individual strategies in performing their tasks, and this would be a significant contribution.

A final, intriguing finding in this manuscript is that animal self-paced training led to much slower learning than "manual" training, by having the experimenter introduce the animal to the apparatus for a few hours each day. Manual training resulted in significantly faster learning, in almost half the number of trials on average, and with significantly fewer omitted trials. This finding does not necessarily argue that manual training is universally a better choice, because it led to more limited water consumption. However, it suggests that there is a distinct contribution of experimenter interactions and/or switching contexts in cognitive training, for example, by activating an "occasion setting" process to accelerate learning for a distinct period of time. Limiting experimenter interactions with mice may be a labor saving intervention, but may not necessarily improve performance. This could be an interesting topic of future investigation, of relevance to understanding how animals of all species learn.

---

## [Author Response]

The following is the authors’ response to the original reviews.

**Reviewer #1 (Public review):**
Summary:This is a new and important system that can efficiently train mice to perform a variety of cognitive tasks in a flexible manner. It is innovative and opens the door to important experiments in the neurobiology of learning and memory.Strengths:Strengths include: high n's, a robust system, task flexibility, comparison of manual-like training vs constant training, circadian analysis, comparison of varying cue types, long-term measurement, and machine teaching.Weaknesses:I find no major problems with this report.Minor weaknesses:(1) Line 219: Water consumption per day remained the same, but number of trails triggered was more as training continued. First, is this related to manual-type training? Also, I'm trying to understand this result quantitatively, since it seems counter-intuitive: I would assume that with more trials, more water would be consumed since accuracy should go up over training (so more water per average trial). Am I understanding this right? Can the authors give more detail or understanding to how more trials can be triggered but no more water is consumed despite training?

Thanks for the comment. We would like to clarify the phenomenon described in Line 219: As the training advanced, the number of trials triggered by mice per day decreased (rather than increased as you mentioned in the comment) gradually for both manual and autonomous groups of mice (Fig. 2H left). The performance, as you mentioned, improved over time (Fig. 2D and 2E), leading to an increased probability of obtaining water and thus relatively stable daily water intake (Fig. 2H middle). We believe the stable daily intake is the minimum amount of water required by the mice under circumstance of autonomous behavioral training. To make the statement more clearly, we indicated the corresponding figure numbers in the text.

Results “… As shown in Fig. 2H, autonomous training yielded significantly higher number of trial/day (980 ± 25 vs. 611 ± 26, Fig. 2H left) and more volume of water consumption/day (1.65 ± 0.06 vs. 0.97 ± 0.03 ml, Fig. 2H middle), which resulted in monotonic increase of body weight that was even comparable to the free water group (Fig.2H right). In contrast, the body weight in manual training group experienced a sharp drop at the beginning of training and was constantly lower than autonomous group throughout the training stage (Fig. 2H right).”

(2) Figure 2J: The X-axis should have some label: at least "training type". Ideally, a legend with colors can be included, although I see the colors elsewhere in the figure. If a legend cannot be added, then the color scheme should be explained in the caption.

Thanks for the suggestion. The labels with corresponding colors for x-axis have been added for Fig. 2J.

(3) Figure 2K: What is the purple line? I encourage a legend here. The same legend could apply to 2J.

Thanks for the suggestion. The legend has been added for Fig. 2K.

(4) Supplementary Figure S2 D: I do not think the phrase "relying on" is correct. Instead, I think "predicted by" or "correlating with" might be better.

We thank the reviewer for the valuable suggestion. The phrase has been changed to ‘predicted by’ for better suitability.

Figure S2 “(D), percentage of trials significantly predicted by different regressors during task learning. …”

**Reviewer #2 (Public review):**
Summary:The manuscript by Yu et al. describes a novel approach for collecting complex and different cognitive phenotypes in individually housed mice in their home cage. The authors report a simple yet elegant design that they developed for assessing a variety of complex and novel behavioral paradigms autonomously in mice.Strengths:The data are strong, the arguments are convincing, and I think the manuscript will be highly cited given the complexity of behavioral phenotypes one can collect using this relatively inexpensive ($100/box) and high throughput procedure (without the need for human interaction). Additionally, the authors include a machine learning algorithm to correct for erroneous strategies that mice develop which is incredibly elegant and important for this approach as mice will develop odd strategies when given complete freedom.Weaknesses:(1) A limitation of this approach is that it requires mice to be individually housed for days to months. This should be discussed in depth.

Thank you for raising this important point. We agree that the requirement for individual housing of mice during the training period is a limitation of our approach, and we appreciate the opportunity to discuss this in more depth. In the manuscript, we add a section to the Discussion to address this limitation, including the potential impact of individual housing on the mice, the rationale for individual housing in our study, and efforts or alternatives made to mitigate the effects of individual housing.

Discussion “… Firstly, our experiments were confined to single-housed mice, which is known to influence murine behavior and physiology, potentially affecting social interaction and stress levels [76]. In our study, individual housing was necessary to ensure precise behavioral tracking, eliminate competitive interactions during task performance, and maintain consistent training schedules without disruptions from cage-mate disturbances. However, the potential of group-housed training has been explored with technologies such as RFID **[**28,29,32–34] to distinguish individual mice, which potentially improving the training efficiency and facilitating research of social behaviors [77]. Notably, it has shown that simultaneous training of group-housed mice, without individual differentiation, can still achieve criterion performance [25].”

(2) A major issue with continuous self-paced tasks such as the autonomous d2AFC used by the authors is that the inter-trial intervals can vary significantly. Mice may do a few trials, lose interest, and disengage from the task for several hours. This is problematic for data analysis that relies on trial duration to be similar between trials (e.g., reinforcement learning algorithms). It would be useful to see the task engagement of the mice across a 24-hour cycle (e.g., trials started, trials finished across a 24-hour period) and approaches for overcoming this issue of varying inter-trial intervals.

Thank you for your insightful comment regarding the variability in inter-trial intervals and its potential impact on data analysis. We agree that this is an important consideration for continuous self-paced tasks.

In our original manuscript, we have showed the general task engagement across 24-hour cycle (Fig. 2K), which revealed two peaks of engagements during the dark cycle with relatively fewer trials during the light cycle. To facilitate analyses requiring consistent trial durations, we defined trial blocks as sequences between two no-response trials. Notably, approximately 66.6% of trials occurred within blocks of >5 consecutive trials (Fig. 2L), which may be particularly suitable for such analyses.

In the revised manuscript, we also added the analysis of the histogram of inter-trial-interval for both the autonomous and manual training paradigms in HABITS (Fig. S2H), which shows that around 55.2% and 77.5% of the intervals are less than 2 seconds in autonomous and manual training, respectively.

Results “… We found more than two-third of the trials was done in >5-trial blocks (Fig. 2L left) which resulted in more than 55% of the trials were with inter-trial-interval less than 2 seconds (Fig. S2H).”

Regarding the approaches to mitigate the issue of varying inter-trial interval, we observed that manual training (i.e., manually transferring to HABITS for ~2 hr/day) in Fig. S2H resulted in more trials with short inter-trial-interval, suggesting that constrained access time promotes task engagement and reduces interval variability. Fig. 2L also indicated that the averaged correct rate increased and the earlylick rate decreased as the length of block increased. This approach could be valuable for studies where consistent trial timing is critical. In the context of our study, we could actually introduce a light, for example, to serve as the cue that prompt the animals to engage during a fixed time duration in a day.

Discussion “… In contrast, the self-paced nature of autonomous training may permit greater variability in attentional engagement 83 and inter-trial-intervals, which could be problematic for data analysis relaying on consistent intervals and/or engagements. Future studies should explore how controlled contextual constraints enhance learning efficiency and whether incorporating such measures into HABITS could optimize its performance.”

(3) Movies - it would be beneficial for the authors to add commentary to the video (hit, miss trials). It was interesting watching the mice but not clear whether they were doing the task correctly or not.

Thanks for the reminder. We have added subtitles to both of the videos. Since the supplementary video1 was not recorded with sound, the correctness of the trials was hard to judge. We replaced the video with another one with clear sound recordings, and the subtitles were commented in detail.

(4) The strength of this paper (from my perspective) is the potential utility it has for other investigators trying to get mice to do behavioral tasks. However, not enough information was provided about the construction of the boxes, interface, and code for running the boxes. If the authors are not willing to provide this information through eLife, GitHub, or their own website then my evaluation of the impact and significance of this paper would go down significantly.

Thanks for this important comment. We would like to clarify that the construction methods, GUI, code for our system, PCB and CAD files (newly uploaded) have already been made publicly available on https://github.com/Yaoyao-Hao/HABITS. Additionally, we have open-sourced all the codes and raw data for all training protocols (https://doi.org/10.6084/m9.figshare.27192897). We will continue to maintain these resources in the future.

Minor concerns:(5) Learning rate is confusing for Figure 3 results as it actually refers to trials to reach the criterion, and not the actual rate of learning (e.g., slope).

Thanks for pointing this out. The ‘learning rate’ which refers to trial number to reach criterion has been changed to ‘the number of trials to reach criterion’.

**Reviewer #3 (Public review):**
Summary:In this set of experiments, the authors describe a novel research tool for studying complex cognitive tasks in mice, the HABITS automated training apparatus, and a novel "machine teaching" approach they use to accelerate training by algorithmically providing trials to animals that provide the most information about the current rule state for a given task.Strengths:There is much to be celebrated in an inexpensively constructed, replicable training environment that can be used with mice, which have rapidly become the model species of choice for understanding the roles of distinct circuits and genetic factors in cognition. Lingering challenges in developing and testing cognitive tasks in mice remain, however, and these are often chalked up to cognitive limitations in the species. The authors' findings, however, suggest that instead, we may need to work creatively to meet mice where they live. In some cases, it may be that mice may require durations of training far longer than laboratories are able to invest with manual training (up to over 100k trials, over months of daily testing) but the tasks are achievable. The "machine teaching" approach further suggests that this duration could be substantially reduced by algorithmically optimizing each trial presented during training to maximize learning.Weaknesses:(1) Cognitive training and testing in rodent models fill a number of roles. Sometimes, investigators are interested in within-subjects questions - querying a specific circuit, genetically defined neuron population, or molecule/drug candidate, by interrogating or manipulating its function in a highly trained animal. In this scenario, a cohort of highly trained animals that have been trained via a method that aims to make their behavior as similar as possible is a strength.However, often investigators are interested in between-subjects questions - querying a source of individual differences that can have long-term and/or developmental impacts, such as sex differences or gene variants. This is likely to often be the case in mouse models especially, because of their genetic tractability. In scenarios where investigators have examined cognitive processes between subjects in mice who vary across these sources of individual difference, the process of learning a task has been repeatedly shown to be different. The authors do not appear to have considered individual differences except perhaps as an obstacle to be overcome.The authors have perhaps shown that their main focus is highly-controlled within-subjects questions, as their dataset is almost exclusively made up of several hundred young adult male mice, with the exception of 6 females in a supplemental figure. It is notable that these female mice do appear to learn the two-alternative forced-choice task somewhat more rapidly than the males in their cohort.

Thank you for your insightful comments and for highlighting the importance of considering both within-subject and between-subject questions in cognitive training and testing in rodent models. We acknowledge that our study primarily focused on highly controlled within-subject questions. However, the datasets we provided did show preliminary evidences for the ‘between-subject’ questions. Key observations include:

The large variability in learning rates among mice observed in Fig. 2I;

The overall learning rate difference between male and female subjects (Fig. 2D vs. Fig. S2G);

The varying nocturnal behavioral patterns (Fig. 2K), etc.

We recognize the value of exploring between-subjects differences in mouse model and discussed more details in the Discussion part.

Discussion “Our study was designed to standardize behavior for the precise interrogation of neural mechanisms, specifically addressing within-subject questions. However, investigators are often interested in between-subject differences—such as sex differences or genetic variants—which can have long-term behavioral and cognitive implications [72,74]. This is particularly relevant in mouse models due to their genetic tractability [75]. Although our primary focus was not on between-subject differences, the dataset we generated provides preliminary evidence for such investigations. Several behavioral readouts revealed individual variability among mice, including large disparities in learning rates across individuals (Fig. 2I), differences in overall learning rates between male and female subjects (Fig. 2D vs. Fig. S2G), variations in nocturnal behavioral patterns (Fig. 2K), etc.”

(2) Considering the implications for mice modeling relevant genetic variants, it is unclear to what extent the training protocols and especially the algorithmic machine teaching approach would be able to inform investigators about the differences between their groups during training. For investigators examining genetic models, it is unclear whether this extensive training experience would mitigate the ability to observe cognitive differences, or select the animals best able to overcome them - eliminating the animals of interest. Likewise, the algorithmic approach aims to mitigate features of training such as side biases, but it is worth noting that the strategic uses of side biases in mice, as in primates, can benefit learning, rather than side biases solely being a problem. However, the investigators may be able to highlight variables selected by the algorithm that are associated with individual strategies in performing their tasks, and this would be a significant contribution.

Thank you for the insightful comments. We acknowledge that the extensive training experience, particularly through the algorithmic machine teaching approach, could potentially influence the ability to observe cognitive differences between groups of mice with relevant genetic variants. However, our study design and findings suggest that this approach can still provide valuable insights into individual differences and strategies used by the animals during training. First, the behavioral readout (including learning rate, engagement pattern, etc.) as mentioned above, could tell certain number of differences among mice. Second, detailed modelling analysis (with logistical regression modelling) could further dissect the strategy that mouse use along the training process (Fig. S2B). We have actually highlighted some variables selected by the regression that are associated with individual strategies in performing their tasks (Fig. S2C) and these strategies could be different between manual and autonomous training groups (Fig. S2D). We included these comments in the Discussion part for further clearance.

Discussion “… Furthermore, a detailed logistic regression analysis dissected the strategies mice employed during training (Fig. S2B). Notably, the regression identified variables associated with individual task-performance strategies (Fig. S2C), which also differed between manually and autonomously trained groups (Fig. S2D). Thus, our system could facilitate high-throughput behavioral studies exploring between-subject differences in the future.”

(3) A final, intriguing finding in this manuscript is that animal self-paced training led to much slower learning than "manual" training, by having the experimenter introduce the animal to the apparatus for a few hours each day. Manual training resulted in significantly faster learning, in almost half the number of trials on average, and with significantly fewer omitted trials. This finding does not necessarily argue that manual training is universally a better choice because it leads to more limited water consumption. However, it suggests that there is a distinct contribution of experimenter interactions and/or switching contexts in cognitive training, for example by activating an "occasion setting" process to accelerate learning for a distinct period of time. Limiting experimenter interactions with mice may be a labor-saving intervention, but may not necessarily improve performance. This could be an interesting topic of future investigation, of relevance to understanding how animals of all species learn.

Thank you for your insightful comments. We agree that the finding that manual training led to significantly faster learning compared to self-paced training is both intriguing and important. One of the possible reasons we think is due to the limited duration of engagement provided by the experimenter in the manual training case, which forced the mice to concentrate more on the trials (thus with fewer omitting trials) than in autonomous training. Your suggestion that experimenter interactions might activate an "occasion setting" process is particularly interesting. In the context of our study, we could actually introduce, for example, a light, serving as the cue that prompt the animals to engage; and when the light is off, the engagement was not accessible any more for the mice to simulate the manual training situation. We agree that this could be an interesting topic for future investigation that might create a more conducive environment for learning, thereby accelerating the learning rate.

Discussion “… Lastly, while HABITS achieves criterion performance in a similar or even shorter overall days compared to manual training, it requires more trials to reach the same learning criterion (Fig. 2G). We hypothesize that this difference in trial efficiency may stem from the constrained engagement duration imposed by the experimenter in manual training, which could compel mice to focus more intensely on task execution, resulting in less trial omissions (Fig. 2F). In contrast, the self-paced nature of autonomous training may permit greater variability in attentional engagement 83 and inter-trial-intervals, which could be problematic for data analysis relaying on consistent intervals and/or engagements. Future studies should explore how controlled contextual constraints enhance learning efficiency and whether incorporating such measures into HABITS could optimize its performance.”

**Reviewer #2 (Recommendations for the authors):**
As I mentioned in the weaknesses, I did not see code or CAD drawings for their home cages and how these interact with a computer.

Thanks for the comment. We would like to clarify that the construction methods, GUI, code for our system, PCB and CAD files (newly uploaded) have already been made publicly available on https://github.com/Yaoyao-Hao/HABITS.